# Width of surface rupture zone for thrust earthquakes. Implications for earthquake fault zoning.

Paolo Boncio[1], Francesca Liberi[1], Martina Caldarella[1], Fiia C. Nurminen[2]

[1]CRUST-DiSPUTer, "G. D'Annunzio" University of Chieti-Pescara, Chieti, I-66100, Italy

[2]Oulu Mining School, University of Oulu, Oulu, FI-90014, Finland

*Correspondence to*: Paolo Boncio (paolo.boncio@unich.it)

**Abstract.** The characteristics of the zones of coseismic surface faulting along thrust faults are analysed in order to define the criteria for zoning the Surface Fault Rupture Hazard (SFRH) along thrust faults. Normal and strike-slip faults have been deeply studied by other authors concerning the SFRH, while thrust faults have not been studied with comparable attention.

Surface faulting data were compiled for 11 well-studied historic thrust earthquakes occurred globally ($5.4 \leq M \leq 7.9$). Several different types of coseismic fault scarps characterise the analysed earthquakes, depending on the topography, fault geometry and near-surface materials (simple and hanging wall collapse scarps; pressure ridges; fold scarps and thrust or pressure ridges with bending-moment or flexural-slip fault ruptures due to large-scale folding). For all the earthquakes, the distance of distributed ruptures from the principal fault rupture (r) and the width of the rupture zone (WRZ) were compiled directly from the literature or measured systematically in GIS-georeferenced published maps.

Overall, surface ruptures can occur up to large distances from the main fault (~2,150 m on the footwall and ~3,100 m on the hanging wall). Most of them occur on the hanging wall, preferentially in the vicinity of the principal fault trace (> 50% at distances <~ 250 m). The widest WRZ are recorded where sympathetic slip (Sy) on distant faults occurs, and/or where bending-moment (B-M) or flexural-slip (F-S) fault ruptures, associated with large-scale folds (hundreds of meters to kilometres in wavelength), are present.

A positive relation between the earthquake magnitude and the total WRZ is evident, while a clear correlation between the vertical displacement on the principal fault and the total WRZ is not found.

The distribution of surface ruptures is fitted with probability density functions, in order to define a criterion to remove outliers (e.g. 90% probability of the cumulative distribution function) and define the zone where the likelihood of having surface ruptures is the highest. This might help in sizing the zones of SFRH during seismic microzonation (SM) mapping.

In order to shape zones of SFRH, a very detailed earthquake geologic study of the fault is necessary (the highest
level of SM, i.e., Level 3 SM according to Italian guidelines). In the absence of such a very detailed study (basic
SM, i.e., Level 1 SM of Italian guidelines) a width of ~840 m (90% probability from "simple thrust" database of
distributed ruptures, excluding B-M, F-S and Sy fault ruptures) is suggested to be sufficiently precautionary. For
more detailed SM, where the fault is carefully mapped, one must consider that the highest SFRH is concentrated
in a narrow zone, ~60 m in width, that should be considered as a fault avoidance zone (more than one third of the
distributed ruptures are expected to occur within this zone).
The fault rupture hazard zones should be asymmetric compared to the trace of the principal fault. The average
footwall to hanging wall ratio (FW: HW) is close to 1:2.
These criteria are applicable to "simple thrust" faults, without considering possible B-M or F-S fault ruptures due
to large-scale folding, and without considering sympathetic slip on distant faults. Areas potentially susceptible to
B-M or F-S fault ruptures should have their own zones of fault rupture hazard that can be defined by detailed
knowledge of the structural setting of the area (shape, wavelength, tightness and lithology of the thrust-related
large-scale folds) and by geomorphic evidence of past secondary faulting. Distant active faults, potentially sus-
ceptible to sympathetic triggering, should be zoned as separate principal faults.
The entire database of distributed ruptures (including B-M, F-S and Sy fault ruptures) can be useful in poorly-
known areas, in order to assess the extent of the area within which potential sources of fault displacement hazard
can be present.
The results from this study and the database made available as supplementary material can be used for improving
the attenuation relationships for distributed faulting, with possible applications in probabilistic studies of fault
displacement hazard.

**Key words**

Fault rupture hazard, thrust earthquakes, earthquake fault zoning.

**1 Introduction**

Coseismic surface ruptures during large earthquakes might produce damage to buildings and facilities located on
or close to the trace of the active seismogenic fault. This is known as Surface Fault Rupture Hazard (SFRH), a
localized hazard that could be avoided if a detailed knowledge of the fault characteristics is achieved. The mitiga-
tion of SFRH can be faced by strategies of fault zoning and avoidance or, alternatively, by (or together with)

probabilistic estimates of fault displacement hazard (e.g. Youngs et al., 2003; Petersen et al., 2011). Both strategies need to employ, as accurately as possible, the location of the active fault trace, the expected displacement on the principal fault (i.e. *principal faulting* in Youngs et al., 2003; see below for the definition), the deformation close to the principal fault, and the distribution of other faulting and fracturing away from it (i.e. *distributed faulting* in Youngs et al., 2003; see below for the definition). While the general geometry and the expected displacement of the principal fault can be obtained through a detailed geological study and the application of empirical relationships (e.g. Wells and Coppersmith, 1994), the occurrence of distributed faulting close to and away from the principal fault rupture is particularly difficult to predict, and only direct observations from well-documented case studies may provide insights on how distributed faulting is expected to occur (e.g. shape and size of rupture zones, attenuation relationships for distributed faulting).

A reference example of fault zoning strategy for mitigating SFRH is the Alquist-Priolo Earthquake Fault Zoning Act (A-P Act), adopted by the state of California (USA) in 1972 (e.g. Bryant and Hart, 2007). The A-P Act defines regulatory zones around active faults (Earthquake Fault Zones, EFZ), within which detailed geologic investigations are required prior to build structures for human occupancy. The boundaries of the EFZ are placed 150-200 m away from the trace of major active faults, or 60 to 90 m away from well-defined minor faults, with exceptions where faults are complex or not vertical. Moreover, the A-P Act defines a minimum distance of 50 feet (15 m) from the well-defined fault trace within which structures designed for human occupancy cannot be built (fault setback), unless proven otherwise. Similarly, the New Zealand guidelines for development of land on or close to active faults (Kerr et al., 2003) define a fault avoidance zone to ensure life safety. Fault avoidance zones on district planning maps will allow a council to restrict development within the fault avoidance zone and take a risk-based approach to development in built-up areas. The risk-based approach combines the key elements of fault recurrence interval, fault complexity and building importance category. The guidelines recommend a minimum buffer of 20 m either sides of the known fault trace (or the likely rupture zone), unless detailed fault studies prove that the deformed zone is less than that.

Recently, in Italy the Department for Civil Protection published guidelines for land management in areas affected by active and capable faults. For the purpose of the guidelines, an active and capable fault is defined as a fault with demonstrated evidence of surface faulting during the last 40,000 years (Technical Commission for Seismic Microzonation, 2015; SM Working Group, 2015). The guidelines are a tool for zoning active and capable faults during seismic microzonation (SM). They also contain a number of recommendations to assist land managers and planners. The fault zones vary at different Levels of SM. In the basic SM (Level 1 SM according to SM Working Group, 2015), the active fault is zoned with a wide Warning Zone that is conceptually equivalent to the EFZ of

the A-P Act. The zone should include all the reasonable inferred fault rupture hazard of both the principal fault
and other secondary faults, and should account for uncertainties in mapping the fault trace. The guidelines rec-
ommend a width of the Warning Zone to be 400 m. Within the Warning Zone, the most detailed level of SM
(Level 3 SM) is recommended; this should be mandatory before new development. Level 3 SM implies a detailed
earthquake geology study of the fault. After completing that study, a new, more accurate fault zoning is achieved.
This includes a 30 m-wide Fault Avoidance Zone around the accurately-defined fault trace. If some uncertainties
persist after Level 3 studies, such as uncertainties about fault trace location or about the possibility of secondary
faulting away from the principal fault, the guidelines suggest the use of a wider zone called Susceptible Zone,
within which development is restricted. Uncertainties within the Susceptible Zone can be reduced by additional
site-specific investigations. The guidelines recommend a width of the Susceptible Zone to be 160 m, but the final
shape and size of the zone depend on the local geology and the level of accuracy reached during Level 3 SM
studies. Both Fault Avoidance and Susceptible Zones can be asymmetric compared with the main fault trace,
with recommended footwall to hanging wall ratios of 1:4, 1:2 and 1:1 for normal, thrust and strike-slip faults, re-
spectively.
Shape and width of the zones in the Italian guidelines are based mostly on data from normal faulting earthquakes
(e.g. Boncio et al., 2012). In general, the fault displacement hazard of normal and strike-slip faults (e.g. Youngs
et al., 2003; Petersen et al., 2011) has been much more studied than that of thrust faults. Zhou et al. (2010) ana-
lysed the width of the surface rupture zones of the 2008 Wenchuan earthquake focusing on the rupture zone close
to the principal fault, with implications on the setback distance. However, to our knowledge, a global data compi-
lation from well-documented surface thrust faulting earthquakes aimed at analysing the characteristics of the
WRZ is lacking in the scientific literature.
The objectives of this work are: 1) to compile data from well-studied surface faulting thrust earthquakes globally
(we analysed 11 earthquakes with magnitudes ranging from 5.4 to 7.9); 2) to analyse statistically the distribution
of surface ruptures compared to the principal fault and the associated WRZ; and 3) to compare the results with
the Italian guidelines and discuss the implications for earthquake fault zoning.

## 2 Methodology

This work analyses the data from 11 well-studied historic surface faulting thrust earthquakes occurred worldwide
during the last few decades (Table 1). These historic earthquakes range in magnitude (Mw) from 5.4 to 7.9 and
belong to different tectonic settings, such as continental collision (Spitak, 1988; Kashmir, 2005; Wenchuan,

2008), fold-and-thrust belt (El Asnam, 1980), oceanic-continental or continental-continental collision in large-scale subduction systems (Chi-Chi, 1999; Nagano, 2014), transform plate boundary (San Fernando, 1971; Coalinga-Nunez, 1983) and intraplate regions (Marryat Creek, 1986; Tennant Creek, 1988; Killari, 1993).

We compiled from the literature data on both principal and distributed faulting, as defined by Youngs et al. (2003). Principal faulting is displacement along the main fault responsible for the release of seismic energy during the earthquake. At the surface, the displacement may occur along a single narrow trace of the principal fault or within a meters-scale wide fault zone. Distributed faulting is displacement on other faults in the vicinity of the principal fault rupture. Distributed ruptures are often discontinuous and may occur tens of meters to kilometers away from the principal fault rupture. Displacement may occur on secondary faults connected with the principal fault, such as splay faults, or on pre-existing faults structurally unconnected with the main fault (called here sympathetic fault ruptures). In particular, for the purpose of this work, the following parameters were extracted from the literature listed in Table 1: i) displacement (vertical, horizontal and net slip, if available) on the principal fault rupture and coordinates of the referred measurement points for strands of the principal fault having associated distributed ruptures; ii) distance from the principal fault to the distributed ruptures (r in Fig. 1), distinguishing between the ones on hanging wall and on footwall; iii) displacement on distributed ruptures (if available); iv) width of the rupture zone (WRZ), distinguishing between the ones on hanging wall and on footwall; and v) scarp type (Fig. 2).

When available, the surface rupture data was compiled directly from published tables (e.g., Chi-Chi, 1999; Wenchuan, 2008), but in most of the other cases the rupture data was measured from the maps published by the previous authors that were GIS-georeferenced for the purpose of this work. Figure 1 displays the technique used for measuring the distance between the principal fault rupture (PF) and the distributed ruptures (DR), which allowed us to sample the rupture zone systematically and in reasonable detail. The measurements carried out on the published maps are illustrated in Fig.s S1 to S11 of the online supplementary material, and the entire compiled database is made available in Table S1. The accuracy of the measurements depends on the scale of the original maps and on the level of detail reported in the maps (the original scale of the published maps is reported in the figures of the supplementary material). In this work only detailed maps were considered, and uncertain or inferred ruptures were not taken into account. It is important to specify that the database made available in Table S1 of the supplementary material can be used only for analysing distributed faulting. Data on the principal fault rupture are not complete, because the strands of the principal fault without distributed ruptures were not considered.

In order to distinguish the principal fault rupture from distributed ruptures, all of the following were considered: 1) larger displacement compared to distributed faulting; 2) longer continuity; 3) coincidence or nearly coinci-

dence with major tectonic/geomorphologic features, such as the trace of the main fault mapped before the earth-
quake on geologic maps.
The distance was measured perpendicularly to the average direction of the principal fault, which was defined by
visual inspection of the published maps, averaging the direction of first-order sections of the principal fault rup-
ture (few to several km-long). Particular attention was paid close to variations of the average strike, in order to
avoid duplicate measurements. In some places, the principal fault rupture is discontinuous. In few of those cases,
and only for the purpose of measuring the distance of distributed ruptures from the main fault trace, we drew the
trace of the main geologic fault between nearby discontinuous ruptures by using major tectonic/geomorphologic
features from available maps (inferred trace of the principal geologic fault in Fig.s S1, S2, S8, S9, S10 and S11).
Distributed ruptures were measured every 200 m along-strike the principal fault. In order to prevent that short
ruptures would be missed or under-sampled during measurement, ruptures shorter than 200 m were measured at
the midpoint, and ruptures between 200 and 400 m-long were measured at the midpoint and endpoints (Fig. 1).
Moreover, all the points having displacement information on distributed ruptures were measured. All the points
with displacement values on the principal fault rupture were also measured if distributed ruptures were associated
with that strand of the principal fault. A particular metrics was used for the Sylmar segment of the San Fernando
1971 rupture zone (Fig. S1) where most of the distributed faulting was mapped along roads, resulting in a very
discontinuous pattern of surface ruptures. In order to have a database of measurements statistically equivalent re-
spect to the other studied earthquakes, variable measurement logics were used in order to sample ruptures at dis-
tances that equal more or less 200 m (see Fig. S1 for details).
All the distributed ruptures reported in the published maps as of primary (i.e., tectonic) origin were measured.
Only the "Beni Rached" rupture zone of the 1981 El Asnam earthquake (Fig. S2) was not measured. It consists of
normal fault ruptures interpreted to be related to either or both (Yelding et al., 1981; Philip and Meghraoui,
1983): 1) very large gravitational sliding; and 2) surface response of an unconstrained deep tectonic fault also re-
sponsible for the 1954 M 6.7 earthquake. Therefore, we avoided measuring the rupture due to the large uncertain-
ties concerning its primary origin.
Some distributed ruptures reasonably unconnected with the main seismogenic fault were classified as sympathet-
ic fault ruptures (Sy; Figs. S1, S2 and S5). We included in this category a rupture on a pre-existing thrust fault
located more than 2 km in the hanging wall of the Chi-Chi 1999 principal fault rupture, due to its large distance
from the main fault trace compared to all the other distributed ruptures (Tsauton East fault, Fig. S8), but a deep
connection with the main seismogenic fault is possible (Ota et al., 2007a).
The measured ruptures have been classified according to the scarp types illustrated in Fig. 2, alternatively the
scarp type was classified as "Unknown". Scarp types from "a" to "g" (Fig. 2) follow the scheme proposed by
Philip et al. (1992), integrated with the classification of Yu et al. (2010). In case of steeply dipping faults, a sim-
ple thrust scarp in bedrock (type a) or a hanging wall collapse scarp in bedrock or in brittle unconsolidated mate-
rial (type b) are produced. In case of low-angle faults and presence of soft-sediment covers, various types of pres-
sure ridges (types c to f) can be observed, depending on the displacement, sense of slip and behaviour of near-
surface materials. In presence of shallow blind faults, a fault-related fold scarp may be formed (type g). Moreo-
ver, in this study two additional structural contexts were distinguished, which are characterized by the occurrence
of bending-moment and flexural-slip fault ruptures (Yeats, 1986), associated with large-scale folds (hundreds of
meters to kilometres in wavelength). Both of these occurred widely during the 1980 El Asnam earthquake (Philip
and Meghraoui, 1983). Bending-moment faults (type h in Fig. 2) are normal faults that are formed close to the
hinge zone of large-scale anticlines (extensional faults at the fold extrados in Philip and Meghraoui, 1983), while
flexural-slip faults (type i) are faults that are formed due to differential slip along bedding planes on the limbs of
a bedrock fold. Bending-moment distributed ruptures associated with small-scale folds (meters to dozens of me-
ters in wavelength), which form at the leading edge of the thrust, belong to scarp types "c" to "g".
**3 Width of the Rupture Zone (WRZ): statistical analysis**
The most impressive and recurrent measured features are ruptures occurring along pre-existing fault traces and on
the hanging wall, as the result of the reactivation of the main thrust at depth. Distributed ruptures are mainly rep-
resented by synthetic and antithetic faults, which are parallel to or branching from the main fault. Fault segmenta-
tion and en échelon geometries are common in transfer zones or in oblique-slip earthquakes.
The collected data was analysed in order to evaluate the width of the rupture zone (WRZ), intended as the total
width, measured perpendicularly to the principal fault rupture, within which all the distributed ruptures occur.
Figure 3 shows frequency distribution histograms of the distance of distributed ruptures from the principal fault
(r) for all the analysed earthquakes. Negative values refer to the footwall, while positive values refer to the hang-
ing wall. In particular, in Fig. 3a we distinguished the scarps with bending-moment (B-M), flexural-slip (F-S) or
sympathetic (Sy) fault ruptures from the other types; in Fig. 3b the scarps without B-M, F-S or Sy fault ruptures
are distinguished by scarp types, and in Fig. 3c the scarps with B-M, F-S or Sy fault ruptures are distinguished by
earthquake. In general, although the values span over a large interval (-2,150 m in the footwall; 3,100 m in the
hanging wall), most of them occur in the proximity of the principal fault and display an asymmetric distribution
between hanging wall and footwall.
In Fig. 3b all the data (excluding scarps with B-M, F-S and Sy fault ruptures) are distinguished by scarp type.
Simple Pressure Ridges with narrow WRZ prevail. Larger WRZ characterizes back-thrust, low-angle and oblique
pressure ridges, implying that the main thrust geometry, the local kinematics and the near-surface rheology have
a significant control in strain partitioning with consequences on the WRZ, as expected.
The occurrence of B-M or F-S fault ruptures is strictly related to the structural setting of the earthquake area. In
particular, B-M fault ruptures, which are related to the presence of large-scale hanging wall anticlines, were
clearly observed in the El Asnam 1980 (Philip and Meghraoui, 1983) and Kashmir 2005 (southern part of central
segment; Kaneda et al., 2008; Sayab and Khan, 2010) earthquakes. A wide extensional zone (1.8 km-long in the
E-W direction; 1.3 km-wide) formed on the eastern hanging wall side of the Sylmar segment of the San Fernando
1971 surface rupture. The interpretation of such an extensional zone is not straightforward. Nevertheless, the
presence of a macro-anticline in the hanging wall of the Sylmar fault is indicated by subsurface data (Mission
Hill anticline; Tsutsumi and Yeats, 1999). Though it is not possible to clearly classify these structures as B-M
faults in strict sense, it seems reasonable to interpret them as generic fold-related secondary extensional faults.
Therefore, they were plotted in Fig.s 3a and 3c together with B-M fault ruptures. F-S fault ruptures were ob-
served on the upright limb of a footwall syncline in the El Asnam 1980 earthquake.
Ruptures close to the main fault (r < 150 m) are due to processes operating in all the scarp types (Fig. 3b), but for
larger distances the distributed faulting can be affected by other processes such as large-scale folding or sympa-
thetic reactivation of pre-existing faults (Fig.s 3a and 3c), contributing significantly in widening the WRZ.
For the analysis of the statistical distribution of "r", the collected data was fitted with a number of probability
density functions by using the commercial software EasyFitProfessional©V.5.6 (http://www.mathwave.com),
which finds the probability distribution that best fits the data and automatically tests the goodness of the fitting.
We decided to analyse both the database without B-M, F-S and Sy fault ruptures (called here "simple thrust" dis-
tributed ruptures; Fig. 4) and the entire database of distributed ruptures without filtering (Fig. 5). The aim is to
analyse separately: 1) distributed ruptures that can be reasonably related only to (or preferentially to) the coseis-
mic propagation to the ground surface of the main fault rupture; they are expected to occur in a rather systematic
way compared to the main fault trace; and 2) distributed ruptures that are affected also by other, non-systematic
structural features, mostly related to large-scale coseismic folding. The hanging wall and footwall data were fitted
separately and the results are synthesized in Fig.s 4 and 5, where the best fitting distribution curves and the cumu-
lative curves are shown.

For "simple thrust" distributed ruptures, the hanging wall data (Figs. 4a and 4b) has a modal value of 7.1 m. The 90% probability (0.9 of the cumulative distribution function, HW90) seems to be a reasonable value to cut off the outliers (flat part of the curves). It corresponds to a distance of ~575 m from the principal fault. From a visual inspection of the histogram (Fig. 4b), there is an evident sharp drop of the data approximately at the 35% probability (HW35), corresponding to a distance of ~40 m from the principal fault. The second sharp drop of the data in the histogram occurs close to the 50% probability (HW50, corresponding to ~80 m from the principal fault). Also the 3rd quartile is shown (HW75), corresponding to a distance of ~260 m from the main fault. The widths of the zones for the different probabilities (90%, 75%, 50% and 35%) are listed in Table 2a.

The footwall data (Figs. 4c and 4d) has a modal value of the best fitting probability density function of 5 m. By applying the same percentiles used for the hanging wall, a 90% cut off (FW90) was found at a distance of ~265 m from the principal fault. The FW75, FW50 and FW35 correspond to distances of ~120 m, ~45 m and ~20 m from the principal fault, respectively (Table 2a). It is worth noticing that also for the footwall the 35% probability corresponds to a sharp drop of the data.

The ratio between the width of the rupture zone on the footwall and the width of the rupture zone on the hanging wall ranges from 1:1.8 to 1:2.2 (Table 2a), and therefore it is always close to 1:2 independently from the used percentile.

The results of the analysis performed on the entire database of distributed ruptures, including also the more complex secondary structures of B-M, F-S and Sy fault ruptures, is illustrated in Fig. 5 and summarized in Table 2b. As expected, the WRZ is significantly larger than for "simple thrust" distributed ruptures. The HW90, HW75 and HW50 correspond to distances of ~1100 m, ~640 m and ~260 m from the principal fault, respectively. For comparison with the "simple thrust" distributed ruptures, also the HW35 was calculated (~130 m), but it does not correspond with a particular drop of the data in the histogram of Fig. 5b. Instead, a sharp drop is visible at a distance of ~40 m from the principal fault, as for the "simple thrust" database. In the footwall, the FW90, FW75 and FW50 correspond to distances of ~720 m, ~330 m and ~125 m from the principal fault, respectively. The FW35 corresponds to a distance of ~65 m, but the sharp drop of the data in the histogram of Fig, 5d is at a distance of ~20 m from the principal fault, as for the "simple thrust" database.

In order to analyse the potential relationships between WRZ and the earthquake size, in Fig. 6 the total width of the rupture zone (WRZ tot = WRZ hanging wall + WRZ footwall) is plotted against Mw (Fig. 6a) and, for the subset of data having displacement information, against the vertical displacement (VD) on the principal fault (Fig. 6b). The vertical displacement measured at the ground surface is highly sensitive to the shallow geometry of the thrust plane. The net displacement along the slip vector is a more appropriate parameter for considering the

size of the displacement at the surface. However, the net displacement is rarely given in the literature, or can be
obtained only by assuming a fault dip, while VD is the most commonly measured parameter. Therefore, we used
VD as a proxy of the amount of surface displacement. In Fig. 6a a positive relation between the total WRZ and
Mw is clear, particularly if sympathetic (Sy) fault ruptures are not considered. In fact, Sy data appear detached
from the other data, suggesting that their occurrence is only partially dependent on the magnitude of the
mainshock. They also depend on the structural features of the area, such as 1) whether or not an active, favoura-
bly-oriented fault is present, and 2) its distance from the main seismogenic source. A correlation between the to-
tal WRZ and VD is not obvious (Fig. 6b). Even for small values of VD ($< 1$ m) the total WRZ can be as wide as
hundreds of meters, but a larger number of displacement data is necessary for drawing convincing conclusions.
**4 Comparison with Italian guidelines and implications for fault zoning during seismic microzonation**
The definition of the WRZ based on the analysis of the data from worldwide thrust earthquakes can support the
evaluation and mitigation of SFRH. The values reported in Table 2 can be used for shaping and sizing fault zones
(e.g. Warning or Susceptible Zones in the Italian guidelines; Earthquake Fault Zones in the A-P Act) and avoid-
ance zones around the trace of active thrust faults (Table 3).
A first question that needs to be answered is which set of data between "simple thrust" distributed ruptures (Fig.
4; Table 2a) and all distributed ruptures (Fig. 5, Table 2b) is the most appropriate to be used for sizing the fault
zones. The answer is not easy and implicates some subjective choices. In Table 3 we suggest using the results
from "simple thrust" distributed ruptures. The results from all distributed ruptures can be used in areas with poor
geologic knowledge, in order to assess the extent of the area within which potential sources of fault displacement
hazard can be present. Our choices result from the following line of reasoning:
1) The data analysed in this work are from brittle rupture of the ground surface. The measured distributed rup-
tures are always associated with surface faulting on the principal fault. Therefore, the results can be used for zon-
ing the hazard deriving from mechanisms connected with the propagation of the rupture on the main fault plane
up to the surface. Deformations associated with blind thrusting are not analysed. Therefore, the results are not
suitable for zoning ductile tectonic deformations associated with blind thrusting (e.g. folding). Clearly, coseismic
folding occurs both during blind thrusting and surface faulting thrusting. Furthermore, brittle surface ruptures and
other ductile deformations can be strictly connected to each other, making difficult to separate the two compo-
nents, but a global analysis of the entire spectrum of permanent tectonic deformation associated to thrust faulting
need additional data not considered here.

2) In most cases, distributed ruptures occur on secondary structures that are small and cannot be recognized before the earthquake, or that only site-specific investigations could distinguish. Fault zones should include the hazard from this kind of ruptures.

3) Some secondary faults connected with the principal fault can be sufficiently large to have their own geologic and geomorphic signature, and can be recognized before the earthquake. Most likely, close to the surface these structures behave similarly to the principal fault, with their own distributed ruptures. Faults with these characteristics should have their own zone, unless they are included in the principal fault zone.

4) Point 3 also applies to distant large active faults that can undergo sympathetic triggering. They should be zoned as separate principal faults. Using Sy fault ruptures for shaping zones of fault rupture hazard would imply distributing the hazard within areas that can be very large (Fig.s 5, 6). The size of the resulting zone would depend mostly on the structural setting of the analysed areas (presence or not of the fault, distance from the seismogenic source) rather than the mechanics which controls distributed faulting in response to principal faulting.

5) B-M and F-S fault ruptures are not always present. Where present, they occur over distances ranging from hundreds of meters to kilometers (Fig. 3c). In any case, B-M and F-S secondary faults are strictly related to the structural setting of the area (large-scale folding; fold shape, wavelength and tightness; stiffness of folded strata). In fact, B-M fault ruptures commonly observed in historical earthquakes are normal faults. B-M normal faults are expected to occur in the shallowest convex (lengthened) layer of the folded anticline. They can occur only where the bending stress is tensional, that is the convex side of the folded layer, preferentially close to the crest of the anticline and parallel to the anticline hinge. F-S faults can rupture the surface where the steeply-dipping limb of a fold is formed by strata of stiff rocks able to slip along bedding planes (e.g. Fig. 2i). Moreover, it is known that coseismic B-M or F-S faults often reactivate pre-existing fault scarps (e.g. Yeats, 1986) which might help in zoning the associated potential fault rupture hazard before the earthquake. Therefore, knowledge of the structural setting of the area can help in identifying zones potentially susceptible to B-M or F-S faulting, which should be zoned as separate sources of fault rupture hazard.

In Table 3, the total WRZ from the present study is compared with the sizes of the zones proposed by the Italian guidelines for SM studies (Technical Commission for Seismic Microzonation, 2015; SM Working Group, 2015). The values reported in Table 3 could be used for integrating the existing criteria. In particular, the total WRZ from "simple thrust" distributed ruptures is used for sizing Warning Zones (Level 1 SM) and Susceptible and Avoidance Zones (Level 3 SM). The total WRZ from all distributed ruptures is suggested to be used for sizing Warning Zones in areas with poor basic geologic knowledge (Level 1 SM).

The first observation is that the FW:HW ratio proposed by the Italian guidelines is supported by the results of this
study (FW:HW ratio close to 1:2).
Assuming that the 90% probability is a reasonable criterion for cutting the outliers from the analysed population,
the resulting total WRZ (HW + FW) for "simple thrust" distributed ruptures is 840 m (560 m on the HW + 280 m
on the FW). This width could be used for zoning all the reasonably inferred fault rupture hazard, from both the
principal fault and distributed ruptures, during basic (Level 1) SM studies, which do not require high-level specif-
ic investigations. The obtained value is significantly different from that recommended by the Italian guidelines
for Level 1 SM (400 m).
A significant difference between our proposal and the Italian guidelines concerns also the width of the zone that
should be avoided, due to the very high likelihood of having surface ruptures. Though the entire rupture zone
could be hundreds of meters wide, more than one third of distributed ruptures are expected to occur within a nar-
row, 60 m-wide zone. As could be expected, only site-specific paleosismologic investigations can quantify the
hazard from surface faulting at a specific site. In the absence of such a detail, and for larger areas (e.g. municipal-
ity scale) the fault avoidance zone should be in the order of 60 m, shaped asymmetrically compared to the trace
of the main fault (40 m on the HW; 20 m on the FW).
In Table 3 a width of 380 m is proposed for the susceptible zone (Level 3 SM). The choice of defining the width
of the zone as the 3rd quartile (3 out of 4 probability that secondary faulting lies within the zone) is rather arbi-
trary. In fact, the width of the susceptible zone should be flexible. Susceptible zones are used only if uncertainties
remain also after high-level seismic microzonation studies, such as uncertainties on the location of the main fault
trace or about the possibility of secondary faulting away from the main fault. Susceptible zones can also be used
for areas where a not better quantifiable distributed faulting might occur, such as in structurally complex zones
(e.g. stepovers between main fault strands).
**5 Conclusions**
The distribution of coseismic surface ruptures (distance of distributed ruptures from the principal fault rupture)
for 11 well-documented historical surface faulting thrust earthquakes ($5.4 \leq M \leq 7.9$) provide constraints on the
general characteristics of the surface rupture zone, with implications for zoning the surface rupture hazard along
active thrust faults.
Distributed ruptures can occur up to large distances from the principal fault (up to ~3,000 m on the hanging wall),
but most of them occur within few dozens of meters from the principal fault. The distribution of secondary rup-
tures is asymmetric, with most of them located on the hanging wall. Coseismic folding of large-scale folds (hun-
dreds of meters to kilometres in wavelength) may produce bending-moment (B-M) or flexural-slip (F-S) fault
ruptures on the hanging wall and footwall, respectively, widening significantly the rupture zone. Additional wid-
ening of the rupture zone can be due to sympathetic slip on distant active faults (Sy fault ruptures).
The distribution of secondary ruptures for "simple thrust" ruptures (without B-M, F-S, and Sy fault ruptures) can
be fitted by a continuous probability density function, of the same form for both the hanging wall and footwall.
This function can be used for removing outliers from the analysed database (e.g. 90% probability) and define cri-
teria for shaping SFRH zones. These zones can be used during seismic microzonation studies and can help in in-
tegrating existing guidelines. More than one third of the ruptures are expected to occur within a zone of ~60 m
wide. This narrow zone could be used for defining the fault avoiding zone during high-level, municipality-scale
seismic microzonation studies (i.e. Level 3 SM according to the Italian guidelines). The average FW:HW ratio of
the WRZ is close to 1:2, independently from the used percentile.
In addition to the expected rupture zone along the trace of the main thrust, zones potentially susceptible to B-M
or F-S secondary faulting can be identified by detailed structural study of the area (shape, wavelength, tightness
and lithology of the thrust-related large-scale folds) and by scrutinize possible geomorphic traces of past second-
ary faulting. Where recognized, these areas should have their own zones of fault rupture hazard.
The analysis of the entire database of distributed ruptures (Fig. 5) indicates significantly larger rupture zones
compared to the database without B-M, F-S and Sy fault ruptures. This is due to the combination of processes re-
lated to the propagation up to the surface of the main fault rupture and other processes associated with large-scale
coseismic folding, as well as triggering of distant faults. These data can be useful in poorly-known areas, in order
to assess the extent of the area within which potential sources of fault displacement hazard can be present.
The results from this study, particularly the function obtained in Fig. 4, can be used for improving the attenuation
relationships for distributed faulting with distance from the principal fault, with possible applications in probabil-
istic studies of fault displacement hazard (e.g., Youngs et al., 2003; Petersen et al., 2011).
**Competing interests**
The authors declare that they have no conflict of interest.

**Acknowledgements**

The project was funded by Department DiSPUTer, "G. D'Annunzio" University of Chieti-Pescara (research funds to P. Boncio).

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

**Table 1. Earthquakes used for analysing the width of the rupture zone (WRZ).**

| Earthquake | Date | Magnitude | Kin. # | SRL* (km) | MD* (m) | Depth (km) | References for earthquake parameters (a) and WRZ calculation (b) |
|---|---|---|---|---|---|---|---|
| 1) San Fernando, CA, USA | 1971.02.09 | $M_s$ 6.5, $M_w$ 6.6 | R-LL | 16 | 2.5 | 8.9 (USGS) | a) 1 <br> b) 2 |
| 2) El Asnam, Algeria | 1980.10.10 | $M_s$ 7.3, $M_w$7.1 | R | 31 | 6.5 | 10 (USGS) | a) 1 <br> b) 3, 4, 5 |
| 3) Coalinga (Nunez), CA, USA | 1983.06.11 | $M_s$5.4, $M_w$ 5.4 | R | 3.3 | 0.64 | 2.0 (USGS) | a) 1 <br> b) 6 |
| 4) Marryat Creek, Australia | 1986.03.30 | $M_s$ 5.8, $M_w$ 5.8 | R-LL | 13 | 1.3 | 3.0 | a) 1, 7 <br> b) 8, 9 |
| 5) Tennant Creek, Australia | 1988.01.22 (3 events) | $M_s$ 6.3, $M_w$ 6.3 <br> $M_s$ 6.4, $M_w$ 6.4 <br> $M_s$ 6.7, $M_w$ 6.6 | R <br> R-LL <br> R | 10.2 <br> 6.7 <br> 16 | 1.3 <br> 1.17 <br> 1.9 | 2.7 <br> 3.0 <br> 4.2 | a) 1, 10 <br> b) 11 |
| 6) Spitak, Armenia | 1988.12.07 | $M_s$ 6.8, $M_w$ 6.8 | R-RL | 25 | 2.0 | 5.0-7.0 | a) 1, 12 <br> b) 13 |
| 7) Killari, India | 1993.09.29 | $M_s$ 6.4, $M_w$ 6.2 | R | 5.5 | 0.5 | 2.6 | a) 14, 15 <br> b) 15, 16 |
| 8) Chi Chi, Taiwan | 1999.09.20 | $M_w$ 7.6 | R-LL | 72 | 12.7 | 8.0 | a) 17, 18 <br> b) 19, 20, 21, 22, 23, 24, 25, 26, 27, 28, 29, 30, 31, 32, 33, 34, 35, 36, 37, 38, 39, 40, 41 |
| 9) Kashmir, Pakistan | 2005.10.08 | $M_w$ 7.6 | R | 70 | 7.05 (v) | <15.0 | a) 42, 43 <br> b) 43, 44 |
| 10) Wenchuan, China | 2008.05.12 | $M_w$ 7.9 | R-RL | 240 | 6.5 (v) 4.9 (h) | 19.0 (USGS) | a) 45 <br> b) 46, 47, 48, 49, 50, 51, 52, 53, 54, 55, 56, 57, 58, 59 |
| 11) Nagano, Japan | 2014.11.22 | $M_w$ 6.2 | R | 9.3 | 1.5 (v) | 4.5 | a) 60, 62 <br> b) 60, 61, 62 |

# Kin. (kinematics): R = reverse, LL = left lateral, RL = right lateral.
* SRL = surface rupture length; MD = maximum displacement (vector sum, unless otherwise specified; v = vertical; h = horizontal).
References: 1 = Wells and Coppersmith, 1994; 2 =U.S. Geological Survey Staff, 1971; 3 =Yelding et al., 1981; 4 =Philip and Meghraoui,
1983; 5 =Meghraoui et al 1988; 6 = Rymer et al. 1990; 7 = Fredrich et al., 1988; 8 = Bowman and Barlow, 1991; 9 = Machette et al.,
1993; 10 = McCaffrey, 1989; 11 = Crone et al., 1992; 12 = Haessler et al. 1992; 13 = Philip et al. 1992; 14 = Lettis et al., 1997; 15 =
Seeber et al. 1996; 16 = Rajendran et al., 1996; 17 = Wesnousky, 2008; 18 = Shin and Teng, 2001; 19 = Kelson et al., 2001; 20 = Kelson
et al., 2003; 21 = Angelier et al., 2003; 22 = Bilham and Yu, 2000; 23 = Chang and Yang, 2004; 24 = Chen et al., 2000; 25 = Chen et al.,
2003; 26 = Faccioli et al., 2008; 27 = Huang et al., 2008; 28 = Huang et al., 2000; 29 = Huang, 2006; 30 = Kawashima, 2002; 31 = Kona-
gai et al., 2006; 32 = Lee and Loh, 2000; 33 = Lee et al., 2001; 34 = Lee and Chan, 2007; 35 = Lee et al., 2003; 36 = Lee et al., 2010; 37 =
Lin, 2000; 38 = Ota et al., 2001; 39 = Ota et al., 2007a; 40 = Ota et al., 2007b; 41 = Central Geological Survey, MOEA at
http://gis.moeacgs.gov.tw/gwh/gsb97-1/sys8/index.cfm; 42 = Avouac et al., 2006; 43 = Kaneda et al., 2008; 44 = Kumahara and Nakata,
2007; 45 = Xu et al., 2009; 46 = Liu-Zeng et al., 2009; 47 = Liu-Zeng et al., 2012; 48 = Yu et al., 2009; 49 = Yu et al., 2010; 50 = Zhou et
al., 2010; 51 = Zhang et al., 2013; 52 = Chen et al., 2008; 53 = Dong et al., 2008a; 54 = Dong et al., 2008b; 55 = Liu-Zeng et al., 2010; 56
= Wang et al., 2010; 57 = Xu et al., 2008; 58 = Zhang et al., 2012; 59 = Zhang et al., 2010; 60 = Okada et al., 2015; 61 = Ishimura et al.,
2015; 62 = Lin et al., 2015.

**Table 2 - Width of the rupture zone (WRZ) on the hanging wall (HW) and on the footwall (FW) and FW to HW ratio for (a) "simple thrust" distributed ruptures (B-M, F-S and Sy excluded) and (b) all distributed ruptures.**

(a)

| Probability[1] | WRZ **HW** | WRZ **FW** | Total WRZ | **FW:HW** |
|---|---|---|---|---|
| 90% | 575 m | 265 m | 840 m | 1:2.2 |
| 75% | 260 m | 120 m | 380 m | 1:2.2 |
| 50% | 80 m | 45 m | 125 m | 1:1.8 |
| 35%[2] | 40 m | 20 m | 60 m | 1:2 |

(b)

| Probability[1] | WRZ **HW** | WRZ **FW** | Total WRZ | **FW:HW** |
|---|---|---|---|---|
| 90% | 1100 m | 720 m | 1820 m | 1:1.5 |
| 75% | 640 m | 330 m | 970 m | 1:1.9 |
| 50% | 260 m | 125 m | 385 m | 1:2.1 |
| 35%[3] | 130 m | 65 m | 195 m | 1:2 |

[1] Probabilities refer to the cumulative distribution functions of Fig.s 4 and 5.
[2] Corresponding to a sharp drop of data in the histograms of Fig. 4, close to the principal fault.
[3] Calculated for comparison with "simple thrust" database, but not corresponding to particular drops of data in the histograms of Fig. 5.

**Table 3 Comparison between fault zone size from Italian guidelines and the Width of the Rupture Zone (WRZ) from**
**the present study (proposal for integrating fault zoning for thrust faults). PF = principal fault rupture; DR = distrib-**
**uted ruptures; SFRH = surface fault rupture hazard.**

| ZONE[1] | Seismic Micro-zonation[2] | Italian guidelines | Proposed widths of zones from total WRZ (from "simple thrust" DR[3]) | Total WRZ from all DR (including B-M, F-S and Sy) | FW:HW[5] |
|---|---|---|---|---|---|
| Warning Zone (*Zona di atten-zione, ZA*) | Basic (Level 1) | 400 m (FW:HW = 1:2) | **> 380 m** (minimum; 75% prob.) to **840 m** (recommended; 90% prob., all the reasonably inferred hazard from PF and DR) | **1800 m** (90% prob., applicable in poorly-known areas for assessing the extent of potential SFRH) | 1:2 |
| Avoidance Zone (*Zona di rispetto*, ZR) | High-level (Level 3) | 30 m (FW:HW = 1:2) | **60 m** (35% prob.[4], very high hazard) | | 1:2 |
| Susceptible Zone (*Zona di suscet-tibilità*, ZS) | High-level (Level 3) | 160 m (FW:HW = 1:2) | **Variable** (depending on the detail of Level 3 MS and structural complexity) **380 m** (in the absence of particular constraints; 75% prob., precautionary) | | 1:2 |

[1] The original names of zones in the Italian guidelines (in Italian) are in italics.
[2] Different levels of Seismic Microzonation refer to SM Working Group (2015).
[3] B-M, F-S and Sy fault ruptures are not included.
[4] Corresponding to a sharp drop of data in the histograms of Fig. 4.
[5] The computed values (Table 2) have been simplified to 1:2.


Figure 1 Sketch synthesizing the methodology used for measur-
ing the "r" and WRZ data. Distance between the principal fault
rupture and distributed rupture is measured along the line per-
pendicular to the auxiliary line indicating the average direction
of the principal fault, always between the faults. Points with dis-
placement values are prioritised at the expense of the 200 m
metrics (the closest measurement point) when reasonable, in
order to avoid over measuring.

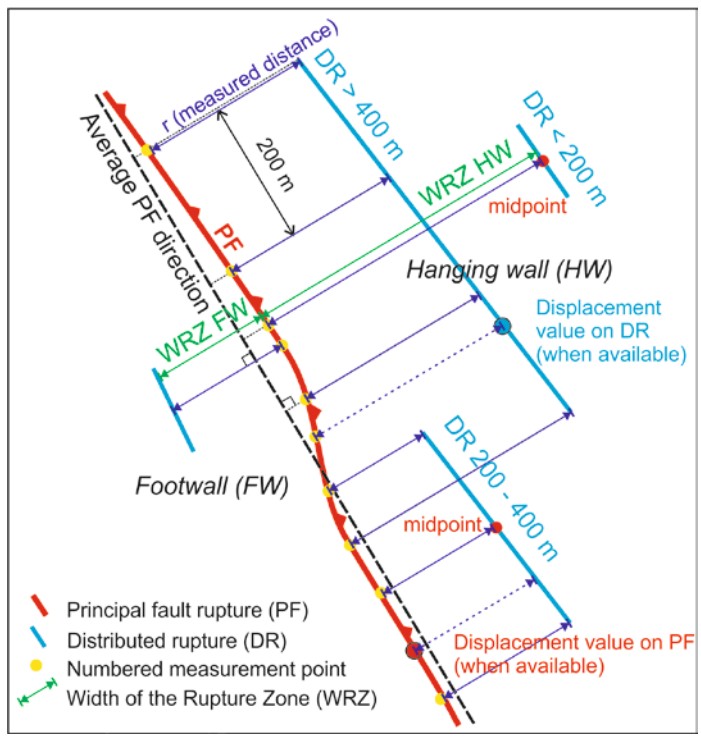


Figure 2 Scarp type classification (modified after Philip et al., 1992 and Yu et al., 2010). The scarp types h) and i) are associated with large-scale folds (hundreds of meters to kilometres in wavelength) and are from Philip and Meghraoui (1983).









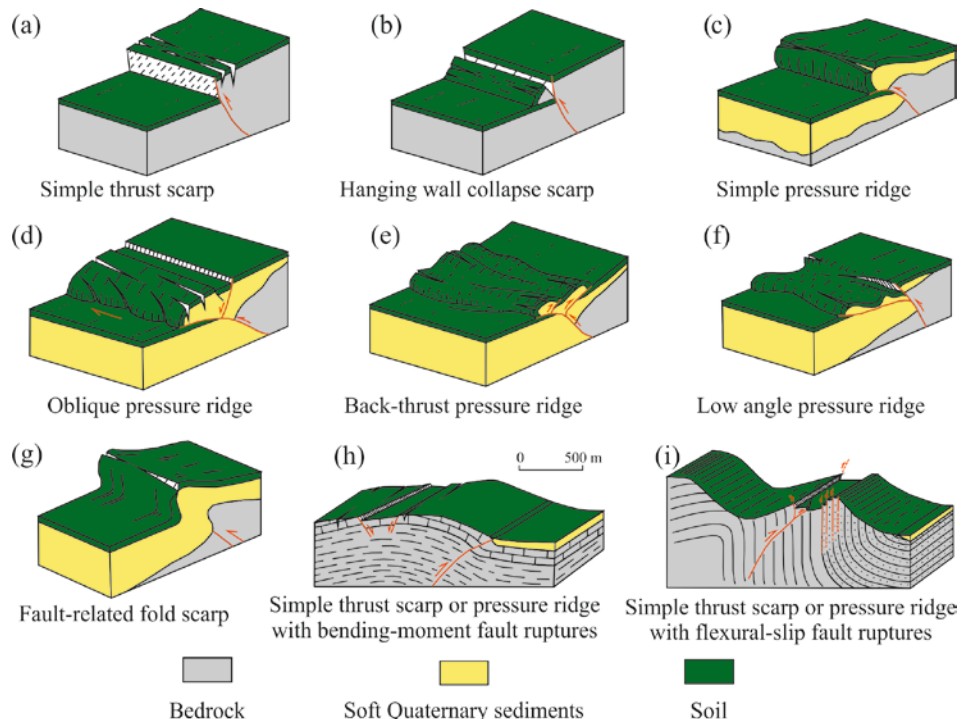

Figure 3 a) Frequency distribu-
tion histogram of the distribut-
ed ruptures distance (r) from
the principal fault rupture (PF)
for the earthquakes reported in
Table 1. The positive and
negative values refer to the
data on the hanging wall and
the footwall, respectively; b)
Frequency distribution curves
of each scarp type excluding
those associated with B-M, F-S
and Sy fault ruptures (types h
and i of Fig. 2 and sympathetic
slip triggered on distant faults);
c) Frequency distribution
curves of the B-M, F-S and Sy
fault ruptures distinguished by
earthquakes (the Sylmar seg-
ment extensional zone of the
San Fernando 1971 earthquake
rupture is included into the B-
M fault ruptures).

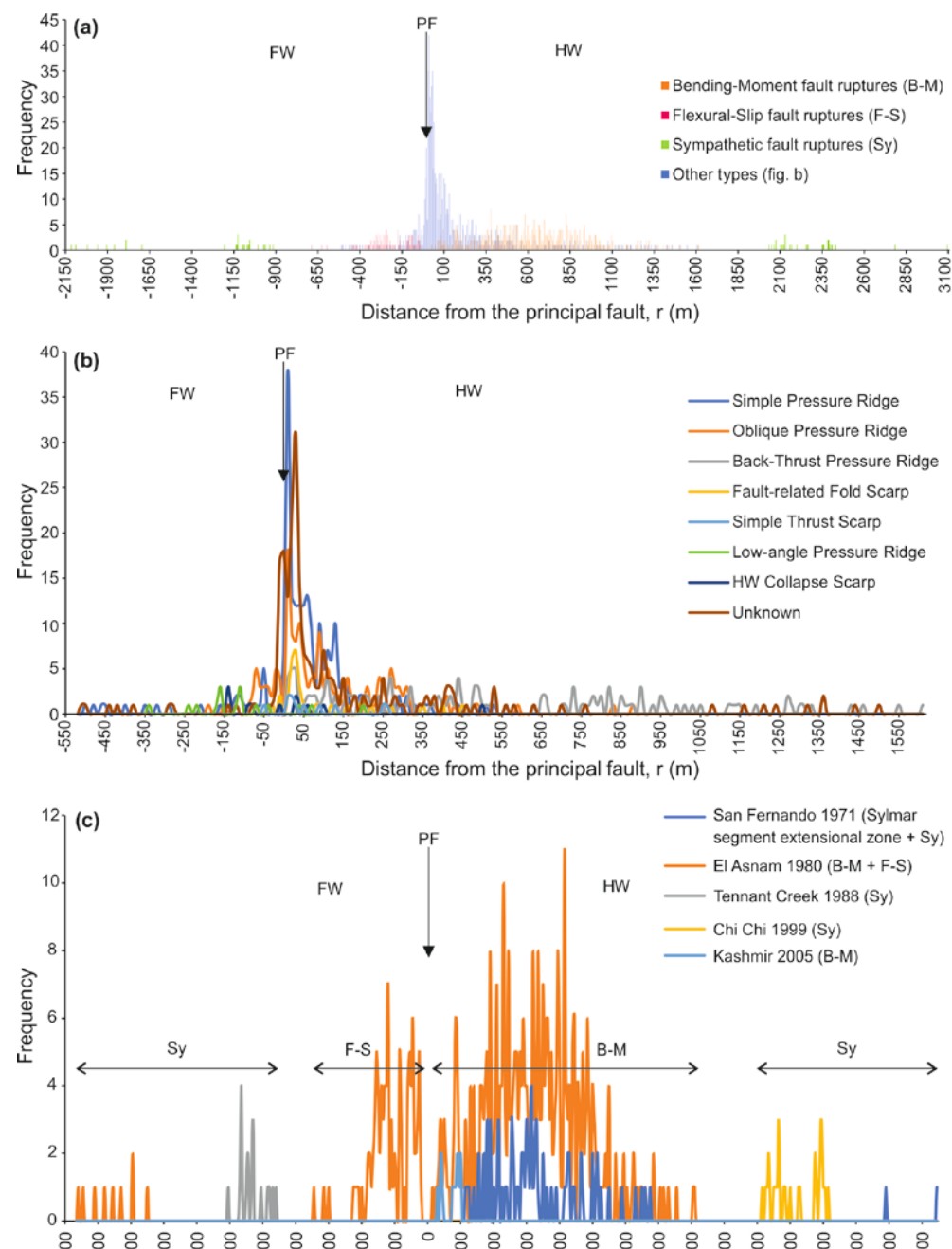


Figure 4 Cumulative distribution function and probability
density function of the rupture distance (r) from the PF for
the hanging wall (a and b, respectively) and the footwall (b
and c, respectively) of the PF. Only the scarp types without
associated B-M, F-S or sympathetic fault ruptures ("simple
thrust" distributed ruptures) were analysed. The 35% prob-
ability (HW35) is indicated because it corresponds to a
sharp drop of the data in the histograms.

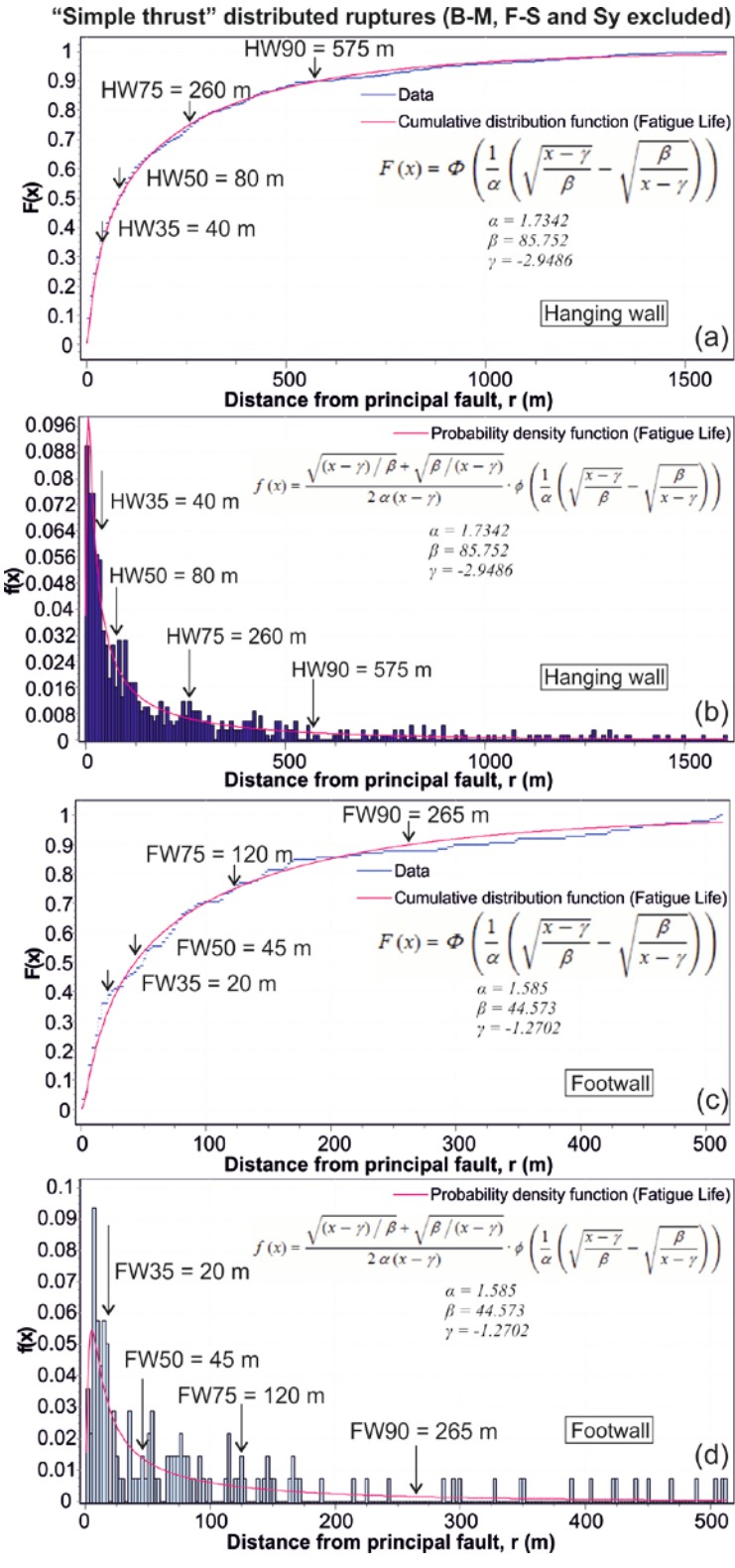

Figure 5 Cumulative distribution function and probability density function of the rupture distance (r) from the PF for the hanging wall (a and b, respectively) and the footwall (c and d, respectively) of the PF. All types of distributed ruptures were considered. The 35% probability (HW35) is indicated for comparison with "simple thrust" database (Fig. 4), but it does not correspond to particular drops of the data in the histograms.

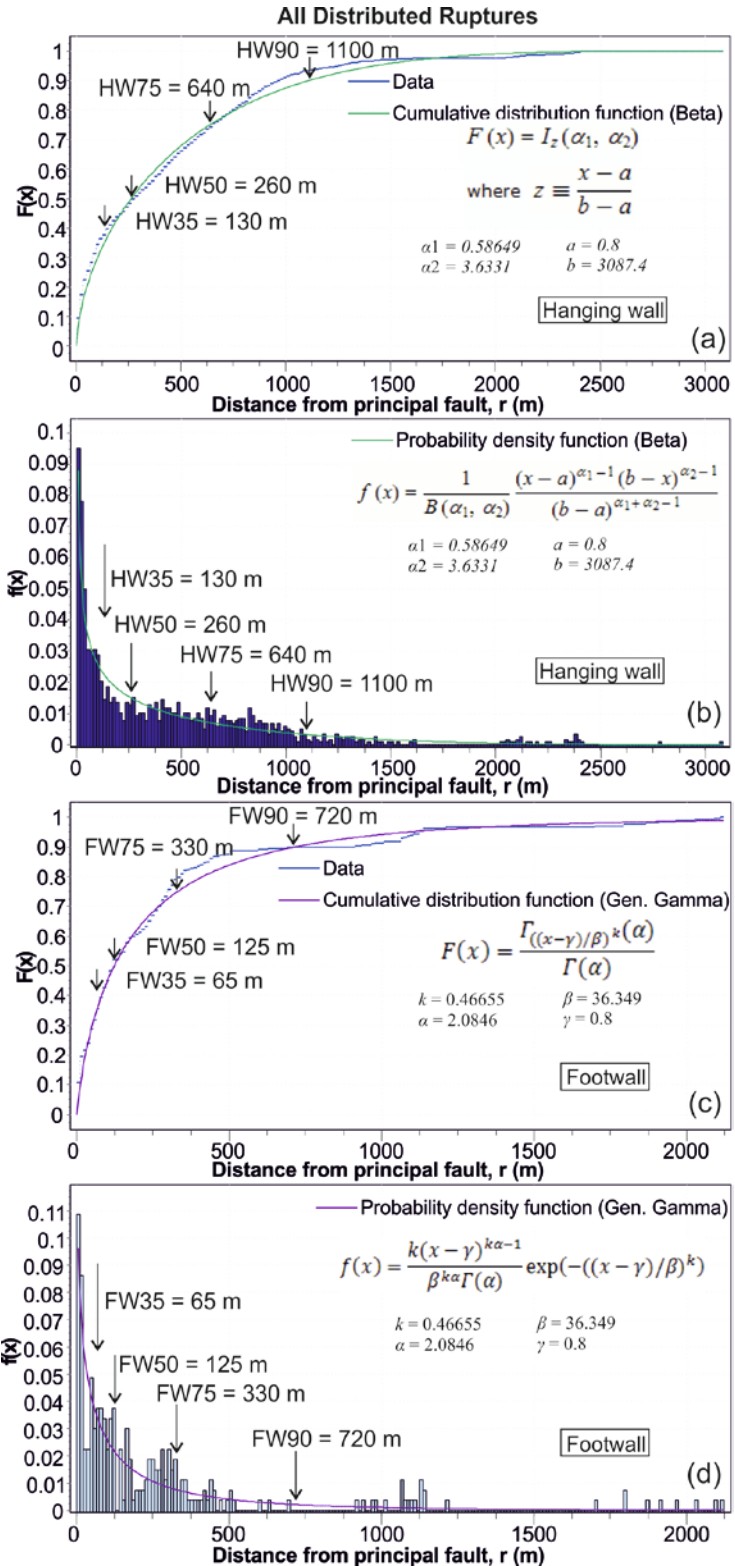


Figure 6 a) Diagram plotting the total
WRZ (WRZtot = WRZ hanging wall +
WRZ footwall) against (a) the earth-
quake magnitude (Mw) and (b) the ver-
tical displacement (VD) on the princi-
pal fault.

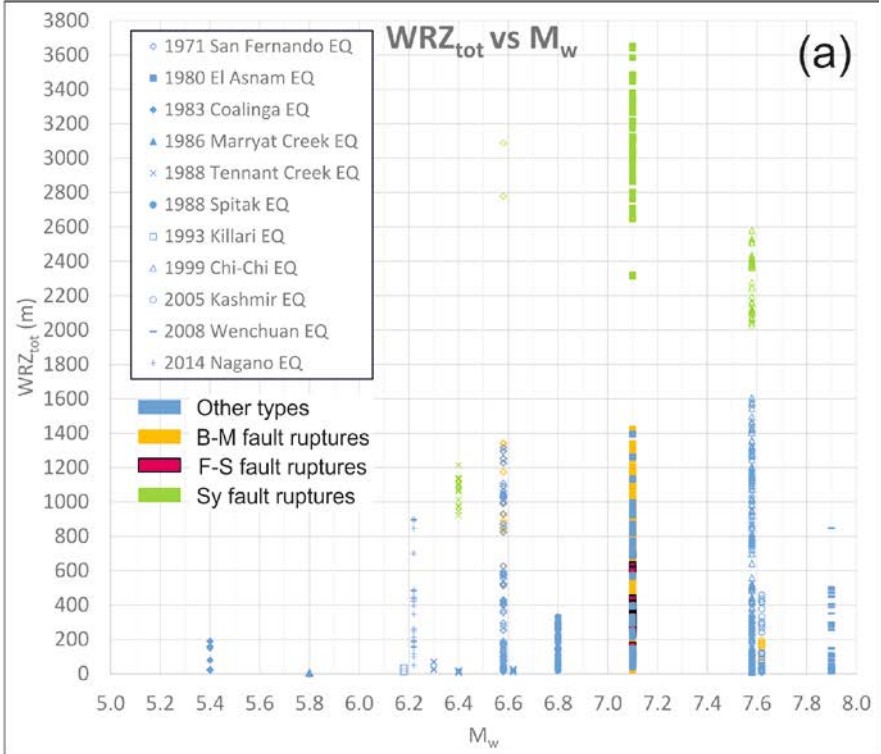

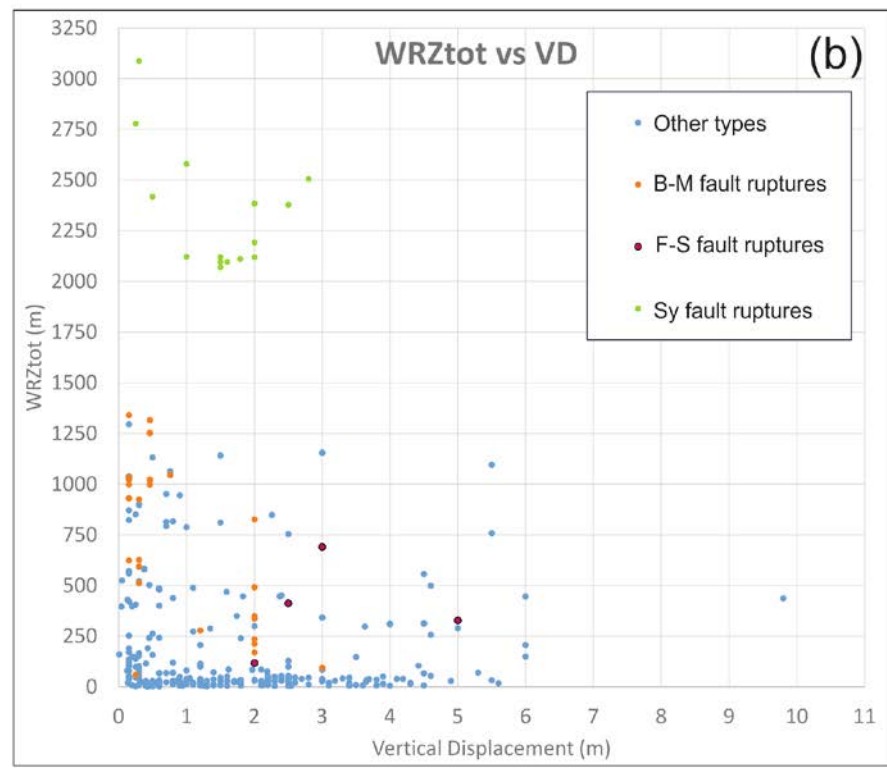