# Peer review of "Width of surface rupture zone for thrust earthquakes. Implications for earthquake fault zoning."

_Natural Hazards and Earth System Sciences, 2017_

## Referee Comment (RC1) · F. Livio (Referee) · 8 May 2017

General comments

Boncio et al. propose a statistical analysis of the surface rupture distribution and an evaluation of the most probable width of surface rupture, occurring during thrust earthquakes. This approach, used for probabilistic fault hazard assessment of distributed faults, is a well-established practice previously performed mainly using strike slip and normal faults datasets. A systematic data review for thrust faults alone was still lacking in literature (partially made on Japanese earthquakes) a thus this work present a novelty aspect that must be considered. The approach is similar to previous studies and no particularly innovations have been proposed from the methodological point of view. An adequate discussion on fold-related faults (i.e., flexural slip faults and bending-moment

faults) has been introduced even if I would have found interesting a statistical approach also on these structures (see below) – even to exclude that a PDF can be invoked for their distribution in space. In summary, I found this work a first interesting (and necessary) review of some well-documented case studies of thrust surface faulting. These observations are resulting in a first proposal of setback distances from fault traces, that should be taken into considerations for siting purposes and by public administrations providing guidelines for land-planning. Neverthelss, some major revisions should be made. In the following, some specific scientific issues are opened to the discussion (Specific Comments) and several notes are made (technical corrections).

Scientific Comments

Firstly, I strongly suggest the Authors to add as supplementary Material the georeferenced maps they used. Ideally, the trace of the main and distributed faults could be provided, as georeferenced shapefiles or .kmz files. This could provide the original datasets that can be used by other scientists for further analysis, data checking etc. and it is one of the main objective of this kind of "data mining" papers. At the moment, no further inspection on the used dataset can be made and this is one of the major faults of the paper in the present form. A note on the methodological approach used for measuring distances. The approach depicted in Figure 1 could result in some biased measurements In fact, it is depending on the azimuth of the main fault strike, in turn derived from the chosen fault tips, fault segmentation etc. This is working well for distributed fault striking parallel to the main one but can be misleading for non-parallel faults. Why not to use a GRID-based approach (like in Petersen al. or in Youngs et al.)? This would also assure data comparison with previous works. At lines 167 – 173 some characteristics of the bending moment faults (BMF), significantly contributing in widening the WRZ, are described. Regarding this point, wavelenght of the thrust-related fold can be considered in order to recognized distant ruptures due to BMF but has not to be taken alone: these secondary structures a more related to hinge zones (and thus geometric characteristics of the fold i.e., curvature of the fold, thickness of the folded

single-layer etc.) than to wavelength alone. Distances proposed to distinguish between pressure ridge anticline vs larger scale structures are just cutoff distance not discussed in their significance. Moreover, I find hard to distinguish between the two of them at intermediate scale. I think that the choice to exclude these structures from a probabilistic analysis can be right but further discussion or objective criteria are needed in order to correctly hierarchize thrust-related faults. Some attempts can be made considering structurally derived cutoff distances: e.g., depth of the sole of the thrust, axial planes (i.e. possible hinge zones) predicted by kink band modeling, etc. In any case, the Authors should provide schematic cross-sections of the considered case studies presenting BMF, so that a direct comparison can be made with the schemes in Figure 2. The best probability density function (PDF) of the distributed faults has been obtained through a commercial software (lines 174-177) but no detailed information is available on the procedure of fitness testing used by the software (a Kolmogorov-Smirnov test is cited but no scores are reported). Maybe this information should be provided as Supplementary Material. A quantitative comparison of the different tested PDF should be provided. Did you test only unimodal distribution or also multimodal? Did you tried to include also bending-moment and flexural-slip faults and fit the entire dataset with a multimodal PDF? Very few people know the Birbaum-Saunders distribution (originally thought to predict the life of mechanical parts subject to stress before failure). Some consideration should be made on the chosen PDF. I found that the statistical analytics are not well explained in the present form of the text and that maybe some other ways of data fitting should have been tested. Lines 185-197 (Figure 4) briefly describe the trend of the fitted PDF considering distances corresponding to progressively increasing cumulative probabilities of occurrence. Here, a strong statistical approach is lacking in transforming cumulative probabilities in distances proposed for setback etc. a qualitative approach is used. The Authors state that "90% probability . . .seems to be a reasonable value to cut the outliers" (line 185-186) and ". . .40% probability bounds reasonably well the zone where the most of the ruptures occur". These statements are not quantitively constrained. If you use a PDF like the Birbaum-Saunders, that can

be characterized by a strong skewness and a long right tail, maybe outliers have to be evaluated with cautiousness. Vanegas, L. H., Rondón, L. M., & Cysneiros, F. J. A. (2012). Diagnostic procedures in Birnbaum–Saunders nonlinear regression models. Computational Statistics & Data Analysis, 56(6), 1662-1680. Provide a review of the tests that can be performed of this PDF in order to identify outliers.

Minor points: Lines 88-93: here, a brief summary of the main pertaining references is given. I suggest to add the following work: - Takao, M., Annaka, T., Kurita, T., 2013. Application of probabilistic fault displacement hazard analysis in Japan. J. Jpn. Assoc. Earthquake Eng. 13 (1), 17–36. Line 249: "and parallel to the anticline hinge"; it depends: not e.g., in transpressive settings. Line 201: "total width": do you mean maximum? Or average? Line 202: did you tried plotting net slip instead of vertical component? Maybe the median of the width could be more clustered. Data on Figure 5 are quite scattered, maybe a bilinear upper bound can be proposed with a flat top toward the right. Line 204-205: also this part is questionable. If we admit that a positive upper bound can be supposed in the lower left of the graph (i.e., less than 200m) how do you explain this threshold distance? Intercept point to ca. 20 m of width, independent from the displacement on the main fault. How do you comment this? It is and expression of aleatory uncertainty or rather related to a geologic process? In any case this result is really important! Line 252: "first order stiffness of the folded material". I don't get the point. What's a "first order stiffness"? are you referring to tensile strength or other mechanical properties of the upper layers? Please, discuss this point or rather avoid this sentence that can be misleading. Line 268-269: "cold criteria" is not appropriate. Do you mean objective? Threshold values?

  Technical corrections

Abstract should be considerably shortened. I would put a major stress on the major advances of this work and novelty, in the first paragraphs. Line 58: put AP Act in refs Line 131: suggested change – "...faults (type i) are reverse faults..." Table 1: indicate also the mapping scale of each digitized map Figure 3: should be a little bit improved

both in the graphing type and in the format. A vectorial image should work best. Figure 4: alpha and beta parameters of the chosen PDF are not discussed in the text or in the caption. Some additional information should be provided and discussed: e.g.., both hangingwall and footwall datasets show similar alpha values but different beta (i.e. median) parameters.

─────────────────────────────

---

## Referee Comment (RC2) · Anonymous Referee #2 · 20 May 2017

General comments

The MS represents a substantial contribution to mitigation of surface faulting hazard, which falls into the scope of NHESS. The paper uses existing worldwide datasets to propose easily applicable criteria to mitigate the surface displacement hazard that can occur during earthquakes. The authors statistically process worldwide data to define "setback" zones as avoidance or warning zones for human occupancy facilities. According to my knowledge, this approach is innovative. Usually, statistics on datasets are used in Fault Displacement Hazard for deriving prediction equations (probability of rupture, attenuation of displacement with distance), but my feeling is that this work is an appropriate and valid way to treat the problem. An interesting outcome is that the statistics tend to confirm part of the Italian regulation lines, but on the other hand the results suggest that the avoidance should be increased for well-mapped cases. I find

the discussion on Bending-Moment and Flexural-Slip ruptures a bit disappointing (see below). Also, the proposed conclusion is more an abstract and I would expect some perspectives to the work that has been done (see below). The MS is clear and concise; contents are well exposed and structured, easy to understand for a wide audience. Figures are good, except the Figure 4 where labels are too small. English is clear to me (English is not my native language). Figures are generally relevant, but I wonder if the Figure 2 (Scarp classification) is really helpful. I suggest to the authors to include in Supplementary Material the rupture maps which would (probably) help the reader to understand some choices about calculation of distances, definition of "average MF direction", etc (see below). The title and abstract clearly and unambiguously reflect the contents of the paper. The authors use an adequate number and quality of accessible references, from which they extract a fair and relevant content. Therefore, I would suggest that Scientific Significance and Quality are good and Presentation Quality is very good (even if some minor corrections would improve the MS). To me, the MS would be accepted after some revision such as follows.

Major issues and recommendations

My first comment concerns the Bending-Moment and Flexural Slip ruptures (distributed deformation features). The authors do not consider these in their analysis because "strictly related to the structural setting of the area (presence and wavelength of the fold". I don't really understand this statement because each rupture, its splays, its pattern of surface deformation is somehow related to a specific structural pattern (geometry of the fault at depth, segmentation, local arrangement of rock packs, etc). Maybe the authors have in mind the fact that BM and FS are rather related to Coseismic folding (ductile deformation) during earthquake than to Coseismic propagation of the rupture plane to the ground surface? I would suggest to the authors to discuss the way these BM and FS distributed ruptures could be accounted for: this is a critical issue for Italy where thrust-related earthquakes usually occur on blind faults and BM and FS on associated fold are the actual main hazard. In section 3, line 173, the authors seem to

mean that there is a direct relation between fold wavelength and location of BM-FS ruptures: this could be a proxy and a way to map and define "Warning or Susceptible zones" to include them in zoning. In line 245-246, the authors write that the "knowledge of the structural setting of the area help in identifying zones potentially susceptible to BM and FS faulting", then why not suggest that such structural features (active folds associated with a thrust) could be defined as "Susceptible Zones"? My second comment deals with the definition of metrics and the chosen hypotheses to calculate distances between secondary ruptures and main ruptures. The Figure 1 presents the approach considering a quite simple case, where the "average MF direction" is easy to infer. It is not clear to me how the authors would cope with curved and/or discontinuous ruptures and scarps; at which scale does the average trace is designed? This would change depending on the rupture size (ex. 240 km of Wenchuan vs 15 km of San Fernando). The MS would largely benefit from the inclusion of the rupture maps, so that the reader would understand the authors' method and eventually reproduce the method to improve their work in further steps. Other questions arise for this point: for instance, how the authors measured the distance of secondary ruptures at the tip of the fault, out of the main trace? Third comment. The results in terms of statistical outcomes are sufficient to support the conclusions (i.e. definition of three different levels of zones). However, I find the section "Conclusions" look like an Abstract. They should include some perspective or prospective insights, like for instance: - How to take the BM and FS ruptures into account? - Do we need more data to build more robust zoning results? - Would this work or compiled database useful in Probabilistic approach of Fault Displacement Hazard Analysis? - Could future similar developments be applied to other tectonic and permanent deformation features like folding, tilting, extensional/compressional strain (see discussions in ANSI/ANS-2.30-2015 Criteria for Assessing Tectonic Surface Fault Rupture and Deformation at Nuclear Facilities)?

Specific points

Replace "associated to" by "associated with" or "related to" Lines 50 51 – Define "main

fault" and "secondary faulting" Line 63 - Ambiguous statement: "The AP act defines a minimum distance (. . .) within which critical facilities and structures designed for human occupancy cannot be built". Delete "Critical facilities" because AP Act is only for housing. Line 68 – Please specify which facilities are concerned by "land management" term: only housing, or also lifelines, pipelines, storages or other facilities Line 73 – The fault zones' widths vary at different Levels instead of The fault zones vary at different Levels Line 82 – Please explain what is decided when the "Susceptible Zone" is defined: Avoidance? Line 88 – Rephrase "In general, worldwide the width of the rupture zone . . ." Lines 92 to 95 – Sentence is to be rephrased. Ambiguous sentence: "1: to collect the data from well-studied (. . .) earthquakes" is not exactly was has been done. Instead, it is a compilation of surface maps and displacement observations, not a collection of data (in the field). Lines 117 to 129 - after reading the whole paper, I am not really sure that this description of scarp classification and related section is useful. What is finally used in the conclusions? The authors may earn space here. Lines 135-136 - clarify whether the scarps are described according to the same classification in the references. Or is it a re-interpretation based on drawings, map? Line 142 – The collected data were instead of The collected data was Lines 154 and 155 – The sentence suggests that there is surface geology information in the references: is that the case or is this an assumption? Line 162 - Subsurface data give indication of the location of an anticline on top of the fault: where are located the FS ruptures with respect to this fold? Lines 168-169 – "the distribution of the BM faults for the El Asnam earthquake is very similar to the distribution of extensional ruptures for the San Fernando earthquake": please provide an explanation to this surprising statement. Is there a similar wavelength of associated fault? So no dependence on magnitude (7.3 against 6.6)? Line 182 – Please explain the Kolmogorov-Smirnov test (if you decide to mention it), its specificity and the reason why it has been selected. Otherwise, you can skip this information. Line 186 – Explain why "the 90% probability (. . .) seems to be a reasonable value to cut the outliers": for statistical reasons considering the examined population? Or because its outcome in terms of setback size fits with an a priori? Line

204 – "the data with BM and FS faults are excluded" instead of "the data with BM and FS faults is excluded" Line 209-210 – "the maximum WRZ, including the secondary ruptures away from the main fault, can be up to 200 m or wider": did you explore if there is a relation between this max. width and the earthquake magnitude? Lines 221 to 225 – I would suggest to moderate this with the conclusion coming from the MB and FS analysis. In cases of verified active folds (related to thrusts), this conclusion should be revised, especially for Italy where most of the thrusts are blind (Po Plain) and could cause folding and BM-FS ruptures at the surface as the major hazard. Line 234 – "Assuming that this relation is robust enough" instead of "Assuming that this relation is real" would be better, I think Line 248 – I was wondering if any BM reverse faulting has never been observed in the synclinal axis. This part requires more perspective. How can we account for these ruptures? Any recommendation? Line 282 – This confirms that the avoidance zone should be larger" would be more appropriate than "This suggests that the avoidance zone should be larger" Table 1 - Why 2014 Nagano earthquake rupture has not been included? Figure 1 - There is a different approach for short, intermediate and long secondary ruptures in measuring distance to main fault. Explain please. Caption - Main fault: How is defined main fault? Max. displacement? On the map, this "main fault" runs along the base of the scarp I presume. In complex cases like (d), (e), (f), Main fault trace is traced along the free face or along the more external topographic bulge? Figure 2 – Explain where the Main fault would be mapped on Figure 1 on each case.

---

## Author Comment (AC1) · 4 Aug 2017

Reply to the comments of Referee #2

We would like to thank the reviewer for the appropriate and very useful comments and suggestions. We are going to comply all the comments in the revised version of the manuscript. The Reviewer's major issues and recommendations have been replied in the following section.

REVIEWER: 1. My first comment concerns the Bending-Moment and Flexural Slip ruptures (distributed deformation features). The authors do not consider these in their analysis because "strictly related to the structural setting of the area (presence and wavelength of the fold". I don't really understand this statement because each rupture,

its splays, its pattern of surface deformation is somehow related to a specific structural pattern (geometry of the fault at depth, segmentation, local arrangement of rock packs, etc). Maybe the authors have in mind the fact that BM and FS are rather related to Co-seismic folding (ductile deformation) during earthquake than to Coseismic propagation of the rup-ture plane to the ground surface? I would suggest to the authors to discuss the way these BM and FS distributed ruptures could be accounted for: this is a critical issue for Italy where thrust-related earthquakes usually occur on blind faults and BM and FS on associated fold are the actual main hazard. In section 3, line 173, the authors seem to mean that there is a direct relation between fold wavelength and location of BM-FS ruptures: this could be a proxy and a way to map and define "Warning or Susceptible zones" to include them in zoning. In line 245-246, the authors write that the "knowledge of the structural setting of the area help in identifying zones potentially susceptible to BM and FS faulting", then why not suggest that such structural features (active folds associated with a thrust) could be defined as "Susceptible Zones"?

RESPONSE: We fully understand the Reviewer's criticism. In fact, the most correct approach for considering secondary faulting (particularly ruptures very far from MF) is a difficult task. Basically, our aim in the paper is to distinguish, if feasible, secondary ruptures that occur in a rather "systematic" way compared to the main fault (i.e., only related to the propagation of the main rupture up to the surface), called "simple thrust ruptures", from secondary ruptures that can be affected by structural features that are not systematic (large-scale folds, lithology of folded rocks, . . .). For sure, we have to discuss this point more deeply and more clearly. We are going to improve this discussion in the revised version. Most importantly, we decided to analyze the data both with and without BMF and F-S secondary ruptures (two different PDFs), and discuss the results and the possible implications. Therefore, we think the paper will be largely improved from this point of view.

REVIEWER: 2. My second comment deals with the definition of metrics and the chosen hypotheses to calculate dis-tances between secondary ruptures and main ruptures. The Figure 1 presents the approach consid-ering a quite simple case, where the "average MF direction" is easy to infer. It is not clear to me how the authors would cope with curved and/or discontinuous ruptures and scarps; at which scale does the average trace is designed? This would change depending on the rupture size (ex. 240 km of Wenchuan vs 15 km of San Fernando). The MS would largely benefit from the in-clusion of the rupture maps, so that the reader would understand the authors' method and eventually reproduce the method to improve their work in further steps. Other questions arise for this point: for instance, how the authors measured the distance of secondary ruptures at the tip of the fault, out of the main trace?

RESPONSE: We will add electronic supplementary material including a summary table with all the measured data and several maps, one for every earthquake, showing the measurement details, including the chosen average strike of the main fault. Moreover, we will explain better the method in the text. In general, we were careful in defining the average strike of the MF and the measurement azimuths, taking into account the variations in strike of the main fault (only first-order, kms-scale strike variations have been considered), and avoiding duplication of measurements. The maps we are adding in the auxiliary material will help the reader in judging our choices. We are also adding data from the Nagano 2014 earthquake, as suggested.

REVIEWER: 3. Third comment. The results in terms of statistical outcomes are suf-ficient to support the conclusions (i.e. definition of three different levels of zones). However, I find the section "Conclusions" look like an Abstract. They should include some perspective or prospective insights, like for instance: - How to take the BM and FS ruptures into account? - Do we need more data to build more robust zon-ing results? – Would this work or compiled database useful in Probabilistic approach of Fault Displacement Hazard Analysis? - Could future similar developments be ap-plied to other tectonic and permanent deformation features like folding, tilting, exten-sional/compressional strain (see discussions in ANSI/ANS-2.30-2015 Criteria for As-sessing Tectonic Surface Fault Rupture and Deformation at Nuclear Facilities)?
RESPONSE: We thank the Reviewer for this good suggestion. This will improve the paper.

RESPONSE to MINOR SPECIFIC POINTS: All the specific points and minor corrections suggested by the Referee will be carefully taken into account during the revision.
* * *

---

## Author Comment (AC2) · 4 Aug 2017

Reply to the comments of Referee #1 (F. Livio)

We would like to thank the reviewer for the appropriate comments and very useful suggestions. We are going to comply all the comments in the revised version of the manuscript. The queries to the scientific comments have been answered separately in the following section.

REVIEWER's SCIENTIFIC COMMENTS: 1) Firstly, I strongly suggest the Authors to add as supplementary Material the georeferenced maps they used. Ideally, the trace of the main and distributed faults could be provided, as georeferenced shapefiles or .kmz files. This could provide the original datasets that can be used by other scientists

for further analysis, data checking etc. and it is one of the main objective of this kind of "data mining" papers. At the moment, no further inspection on the used dataset can be made and this is one of the major faults of the paper in the present form.

RESPONSE: We will add electronic supplementary material including a summary table with all the measured data and several maps, one for every earthquake, showing the measurement details, including the chosen average strike of the main fault.

REVIEWER: 2) A note on the methodological approach used for measuring distances. The approach depicted in Figure 1 could result in some biased measurements In fact, it is depending on the azimuth of the main fault strike, in turn derived from the chosen fault tips, fault segmentation etc. This is working well for distributed fault striking parallel to the main one but can be misleading for non-parallel faults. Why not to use a GRID-based approach (like in Petersen al. or in Youngs et al.)? This would also assure data comparison with previous works.

RESPONSE: Actually, our approach is very similar to that used by Petersen et al. (2011), but more detailed (we used more closely spaced measurements). We will explain better the method in the text. In any case, independently from the used method, we need to define the azimuth along which the distance from the main fault is measured. One target of the paper is zoning the hazard around a mapped fault. Therefore, we need distances from the fault trace. We were careful in defining the measurement azimuth, taking into account the variations in strike of the main fault, and avoiding duplication of measurements. The maps we are adding in the auxiliary material will help the reader in judging our choices.

REVIEWER: 3) At lines 167 – 173 some characteristics of the bending moment faults (BMF), significantly contributing in widening the WRZ, are described. Regarding this point, wavelenght of the thrust-related fold can be considered in order to recognized distant ruptures due to BMF but has not to be taken alone: these secondary structures a more related to hinge zones (and thus geometric characteristics of the fold i.e., curvature of the fold, thickness of the folded single-layer etc.) than to wavelength alone. Distances proposed to distinguish between pressure ridge anticline vs larger scale structures are just cutoff distance not discussed in their significance. Moreover, I find hard to distinguish between the two of them at intermediate scale. I think that the choice to exclude these structures from a probabilistic analysis can be right but further discussion or objective criteria are needed in order to correctly hierarchize thrust-related faults. Some attempts can be made considering structurally derived cutoff distances: e.g., depth of the sole of the thrust, axial planes (i.e. possible hinge zones) predicted by kink band modeling, etc. In any case, the Authors should provide schematic cross-sections of the considered case studies presenting BMF, so that a direct comparison can be made with the schemes in Figure 2.

RESPONSE: We are now analyzing the data both with and without BMF and F-S secondary ruptures (two different PDFs). In general, we agree that all the points suggested by the Referee should be better dis-cussed in the revised version of the manuscript.

REVIEWER: 4) The best probability density function (PDF) of the distributed faults has been obtained through a commercial software (lines 174-177) but no detailed information is available on the procedure of fitness testing used by the software (a Kolmogorov-Smirnov test is cited but no scores are reported). Maybe this information should be provided as Supplementary Material. A quantitative comparison of the different tested PDF should be provided. Did you test only unimodal distribution or also multimodal? Did you tried to include also bending-moment and flexural-slip faults and fit the entire dataset with a multimodal PDF? Very few people know the Birbaum-Saunders distribution (originally thought to predict the life of mechanical parts subject to stress before failure). Some consideration should be made on the chosen PDF. I found that the statistical analytics are not well explained in the present form of the text and that maybe some other ways of data fitting should have been tested. Lines 185-197 (Figure 4) briefly describe the trend of the fitted PDF considering distances corresponding to progressively increasing cumulative probabilities of occurrence. Here, a strong statistical

approach is lacking in transforming cumulative probabilities in distances proposed for setback etc. a qualitative approach is used. The Authors state that "90% probability : : :seems to be a reasonable value to cut the outliers" (line 185-186) and ": : :40% probability bounds reasonably well the zone where the most of the ruptures occur". These statements are not quantitively constrained. If you use a PDF like the Birbaum-Saunders, that can be characterized by a strong skewness and a long right tail, maybe outliers have to be evaluated with cautiousness. Vanegas, L. H., Rondón, L. M., & Cysneiros, F. J. A. (2012). Diagnostic procedures in Birnbaum–Saunders nonlinear regression models. Computational Statistics & Data Analysis, 56(6), 1662-1680. Provide a review of the tests that can be performed of this PDF in order to identify outliers.

RESPONSE: The final PDFs are subject to variations as we are adding additional data (suggestion from Referee 2) and we are analyzing data with and without secondary ruptures belonging to BMF and F-S faults. In general, the aim is to find a PDF only based on its ability to fit the data. In order to find this PDF in the easiest way the possible, we decided to use a commercial software, assuming that the software is working sufficiently well. We think that a deep statistical analysis of the data is very interesting, but beyond the aim of this paper. In any case, all the suggestions by the Referee will be taken into account, including a reading of the suggested reference. Concerning the percentiles used for sizing the zones, we acknowledge that they are subjective choices. In the revised version we will state even more clearly that these are subjective choices. We think that it is very difficult to define really objective criteria. We also think that the reader can accept our suggestions as an "expert judgement" or, most importantly, can make its own choice.

RESPONSE to MINOR POINTS AND TECHNICAL CORRECTIONS: All the minor points and technical corrections suggested by the Referee will be carefully taken into account.

---

## Author Response (AR1)

**Reply to the comments of Referee #1 (F. Livio)**

*REVIEWER:*

*General comments*

*Boncio et al. propose a statistical analysis of the surface rupture distribution and an evaluation of the most probable width of surface rupture, occurring during thrust earthquakes. This approach, used for probabilistic fault hazard assessment of distributed faults, is a well-established practice previously performed mainly using strike slip and normal faults datasets. A systematic data review for thrust faults alone was still lacking in literature (partially made on Japanese earthquakes) a thus this work present a novelty aspect that must be considered. The approach is similar to previous studies and no particularly innovations have been proposed from the methodological point of view. An adequate discussion on fold-related faults (i.e., flexural slip faults and bending-moment faults) has been introduced even if I would have found interesting a statistical approach also on these structures (see below) – even to exclude that a PDF can be invoked for their distribution in space. In summary, I found this work a first interesting (and necessary) review of some well-documented case studies of thrust surface faulting. These observations are resulting in a first proposal of setback distances from fault traces, that should be taken into considerations for siting purposes and by public administrations providing guidelines for land-planning. Nevertheless, some major revisions should be made. In the following, some specific scientific issues are opened to the discussion (Specific Comments) and several notes are made (technical corrections).*

*Scientific Comments*

1) *Firstly, I strongly suggest the Authors to add as supplementary Material the georeferenced maps they used. Ideally, the trace of the main and distributed faults could be provided, as georeferenced shapefiles or .kmz files. This could provide the original datasets that can be used by other scientists for further analysis, data checking etc. and it is one of the main objective of this kind of "data mining" papers. At the moment, no further inspection on the used dataset can be made and this is one of the major faults of the paper in the present form.*

2) *A note on the methodological approach used for measuring distances. The approach depicted in Figure 1 could result in some biased measurements In fact, it is depending on the azimuth of the main fault strike, in turn derived from the chosen fault tips, fault segmentation etc. This is working well for distributed fault striking parallel to the main one but can be misleading for non-parallel faults. Why not to use a GRID-based approach (like in Petersen al. or in Youngs et al.)? This would also assure data comparison with previous works.*

3) *At lines 167 – 173 some characteristics of the bending moment faults (BMF), significantly contributing in widening the WRZ, are described. Regarding this point, wavelenght of the thrust-related fold can be considered in order to recognized distant ruptures due to BMF but has not to be taken alone: these secondary structures a more related to hinge zones (and thus geometric characteristics of the fold i.e., curvature of the fold, thickness of the folded single-layer etc.) than to wavelength alone. Distances proposed to distinguish between pressure ridge anticline vs larger scale structures are just cutoff distance not discussed in their significance. Moreover, I find hard to distinguish between the two of them at intermediate scale. I think that the choice to exclude these structures from a probabilistic analysis can be right but further discussion or objective criteria are needed in order to correctly hierarchize thrust-related faults. Some attempts can be made considering structurally derived cutoff distances: e.g., depth of the sole of the thrust, axial planes (i.e. possible hinge zones) predicted by kink band modeling, etc. In any case, the Authors*

*should provide schematic cross-sections of the considered case studies presenting BMF, so that a direct comparison can be made with the schemes in Figure 2.*

4) *The best probability density function (PDF) of the distributed faults has been obtained through a commercial software (lines 174-177) but no detailed information is available on the procedure of fitness testing used by the software (a Kolmogorov-Smirnov test is cited but no scores are reported). Maybe this information should be provided as Supplementary Material. A quantitative comparison of the different tested PDF should be provided. Did you test only unimodal distribution or also multimodal? Did you tried to include also bending-moment and flexural-slip faults and fit the entire dataset with a multimodal PDF? Very few people know the Birbaum-Saunders distribution (originally thought to predict the life of mechanical parts subject to stress before failure). Some consideration should be made on the chosen PDF. I found that the statistical analytics are not well explained in the present form of the text and that maybe some other ways of data fitting should have been tested. Lines 185-197 (Figure 4) briefly describe the trend of the fitted PDF considering distances corresponding to progressively increasing cumulative probabilities of occurrence. Here, a strong statistical approach is lacking in transforming cumulative probabilities in distances proposed for setback etc. a qualitative approach is used. The Authors state that "90% probability : : :seems to be a reasonable value to cut the outliers" (line 185-186) and ": : :40% probability bounds reasonably well the zone where the most of the ruptures occur". These statements are not quantitively constrained. If you use a PDF like the Birbaum-Saunders, that can be characterized by a strong skewness and a long right tail, maybe outliers have to be evaluated with cautiousness. Vanegas, L. H., Rondón, L. M., & Cysneiros, F. J. A. (2012). Diagnostic procedures in Birnbaum–Saunders nonlinear regression models. Computational Statistics & Data Analysis, 56(6), 1662-1680. Provide a review of the tests that can be performed of this PDF in order to identify outliers.*

*MINOR POINTS*
*Lines 88-93: here, a brief summary of the main pertaining references is given. I suggest to add the following work: - Takao, M., Annaka, T., Kurita, T., 2013. Application of probabilistic fault displacement hazard analysis in Japan. J. Jpn. Assoc. Earthquake Eng. 13 (1), 17–36.*
*Line 249: "and parallel to the anticline hinge"; it depends: not e.g., in transpressive settings.*
*Line 201: "total width": do you mean maximum? Or average?*
*Line 202: did you tried plotting net slip instead of vertical component? Maybe the median of the width could be more clustered. Data on Figure 5 are quite scattered, maybe a bilinear upper bound can be proposed with a flat top toward the right.*
*Line 204-205: also this part is questionable. If we admit that a positive upper bound can be supposed in the lower left of the graph (i.e., less than 200 m) how do you explain this threshold distance? Intercept point to ca. 20 m of width, independent from the displacement on the main fault. How do you comment this? It is and expression of aleatory uncertainty or rather related to a geologic process? In any case this result is really important!*
*Line 252: "first order stiffness of the folded material". I don't get the point. What's a "first order stiffness"? are you referring to tensile strength or other mechanical properties of the upper layers? Please, discuss this point or rather avoid this sentence that can be misleading.*
*Line 268-269: "cold criteria" is not appropriate. Do you mean objective? Threshold values?*

*TECHNICAL CORRECTIONS*

*Abstract should be considerably shortened. I would put a major stress on the major advances of this work and novelty, in the first paragraphs.*

*Line 58: put AP Act in refs*

*Line 131: suggested change – ": : :faults (type i) are reverse faults: : :" Table 1: indicate also the mapping scale of each digitized ma*

*Figure 3: should be a little bit improved both in the graphing type and in the format. A vectorial image should work best.*

*Figure 4: alpha and beta parameters of the chosen PDF are not discussed in the text or in the caption. Some additional information should be provided and discussed: e.g.., both hangingwall and footwall datasets show similar alpha values but different beta (i.e. median) parameters.*

RESPONSE:

We would like to thank the reviewer for the appropriate comments and very useful suggestions. We considered all the comments in the revised version of the manuscript.

The queries to the scientific comments have been answered separately in the following section.

All the minor points and technical corrections suggested by the Referee were taken into account.

*SCIENTIFIC COMMENTS*

1) *Firstly, I strongly suggest the Authors to add as supplementary Material the georeferenced maps they used. Ideally, the trace of the main and distributed faults could be provided, as georeferenced shapefiles or .kmz files. This could provide the original datasets that can be used by other scientists for further analysis, data checking etc. and it is one of the main objective of this kind of "data mining" papers. At the moment, no further inspection on the used dataset can be made and this is one of the major faults of the paper in the present form.*

RESPONSE: We added electronic supplementary material, consisting in a summary table with all the measured data and several maps, one for every earthquake, showing the measurement details and the chosen average strike of the main fault.

2) *A note on the methodological approach used for measuring distances. The approach depicted in Figure 1 could result in some biased measurements In fact, it is depending on the azimuth of the main fault strike, in turn derived from the chosen fault tips, fault segmentation etc. This is working well for distributed fault striking parallel to the main one but can be misleading for non-parallel faults. Why not to use a GRID-based approach (like in Petersen al. or in Youngs et al.)? This would also assure data comparison with previous works.*

RESPONSE: We think our approach is more detailed than that used by Petersen et al. (2011) (we used closely spaced measurements). We will explain better the method in the text.

In any case, independently from the used method, we need to define the azimuth along which the distance from the main fault is measured. One target of the paper is zoning the hazard around a mapped fault. Therefore, we need distances from the fault trace. We were careful in defining the measurement azimuth, taking into account the variations in strike of the main fault, and avoiding duplication of measurements. The maps added in the auxiliary material will help the reader in judging our choices.

3) *At lines 167 – 173 some characteristics of the bending moment faults (BMF), significantly contributing in widening the WRZ, are described. Regarding this point, wavelenght of the thrust-related fold can be considered in order to recognized distant ruptures due to BMF but has not to be taken alone: these secondary structures a more related to hinge zones (and thus geometric characteristics of the fold i.e., curvature of the fold, thickness of the folded single-layer etc.) than to wavelength alone. Distances proposed to distinguish between pressure ridge anticline vs larger scale structures are just cutoff distance not discussed in their significance. Moreover, I find hard to distinguish between the two of them at intermediate scale. I think that the choice to exclude these structures from a probabilistic analysis can be right but further discussion or objective criteria are needed in order to correctly hierarchize thrust-related faults. Some attempts can be made considering structurally derived cutoff distances: e.g., depth of the sole of the thrust, axial planes (i.e. possible hinge zones) predicted by kink band modeling, etc. In any case, the Authors should provide schematic cross-sections of the considered case studies presenting BMF, so that a direct comparison can be made with the schemes in Figure 2.*

RESPONSE: We analyzed the data both with and without BMF and F-S secondary ruptures (two different PDFs). We also differentiated Sympathetic ruptures (Sy). In general, in the revised version of the manuscript we tried to discuss more clearly all the points suggested by the Referee.

4) *The best probability density function (PDF) of the distributed faults has been obtained through a commercial software (lines 174-177) but no detailed information is available on the procedure of fitness testing used by the software (a Kolmogorov-Smirnov test is cited but no scores are reported). Maybe this information should be provided as Supplementary Material. A quantitative comparison of the different tested PDF should be provided. Did you test only unimodal distribution or also multimodal? Did you tried to include also bending-moment and flexural-slip faults and fit the entire dataset with a multimodal PDF? Very few people know the Birbaum-Saunders distribution (originally thought to predict the life of mechanical parts subject to stress before failure). Some consideration should be made on the chosen PDF. I found that the statistical analytics are not well explained in the present form of the text and that maybe some other ways of data fitting should have been tested. Lines 185-197 (Figure 4) briefly describe the trend of the fitted PDF considering distances corresponding to progressively increasing cumulative probabilities of occurrence. Here, a strong statistical approach is lacking in transforming cumulative probabilities in distances proposed for setback etc. a qualitative approach is used. The Authors state that "90% probability : : :seems to be a reasonable value to cut the outliers" (line 185-186) and ": : :40% probability bounds reasonably well the zone where the most of the ruptures occur". These statements are not quantitively constrained. If you use a PDF like the Birbaum-Saunders, that can be characterized by a strong skewness and a long right tail, maybe outliers have to be evaluated with cautiousness. Vanegas, L. H., Rondón, L. M., & Cysneiros, F. J. A. (2012). Diagnostic procedures in Birnbaum–Saunders nonlinear regression models. Computational Statistics & Data Analysis, 56(6), 1662-1680. Provide a review of the tests that can be performed of this PDF in order to identify outliers.*

RESPONSE: The aim is to find PDFs only based on their ability to fit the data. In order to find these PDFs in the easiest way the possible, we decided to use a commercial software, assuming that the software is working sufficiently well. We think that a deep statistical analysis of the data is very interesting, but beyond the aim of this paper.

Concerning the percentiles used for sizing the zones, we acknowledge that they are subjective choices. In the revised version we stated more clearly that these are subjective choices. We think that it is very difficult to define really objective criteria. We also think that the reader can accept our suggestions as an "expert judgement" or, most importantly, can make its own choice.

**Reply to the comments of Referee #2**

*REVIEWER*

*General comments*

*The MS represents a substantial contribution to mitigation of surface faulting hazard, which falls into the scope of NHESS. The paper uses existing worldwide datasets to propose easily applicable criteria to mitigate the surface displacement hazard that can occur during earthquakes. The authors statistically process worldwide data to define "setback" zones as avoidance or warning zones for human occupancy facilities. According to my knowledge, this approach is innovative. Usually, statistics on datasets are used in Fault Displacement Hazard for deriving prediction equations (probability of rupture, attenuation of displacement with distance), but my feeling is that this work is an appropriate and valid way to treat the problem. An interesting outcome is that the statistics tend to confirm part of the Italian regulation lines, but on the other hand the results suggest that the avoidance should be increased for well-mapped cases. I find the discussion on Bending-Moment and Flexural-Slip ruptures a bit disappointing (see below). Also, the proposed conclusion is more an abstract and I would expect some perspectives to the work that has been done (see below). The MS is clear and concise; contents are well exposed and structured, easy to understand for a wide audience. Figures are good, except the Figure 4 where labels are too small. English is clear to me (English is not my native language). Figures are generally relevant, but I wonder if the Figure 2 (Scarp classification) is really helpful. I suggest to the authors to include in Supplementary Material the rupture maps which would (probably) help the reader to understand some choices about calculation of distances, definition of "average MF direction", etc (see below). The title and abstract clearly and unambiguously reflect the contents of the paper. The authors use an adequate number and quality of accessible references, from which they extract a fair and relevant content. Therefore, I would suggest that Scientific Significance and Quality are good and Presentation Quality is very good (even if some minor corrections would improve the MS). To me, the MS would be accepted after some revision such as follows.*

*MAJOR ISSUES AND RECOMMENDATIONS*

1. *My first comment concerns the Bending-Moment and Flexural Slip ruptures (distributed deformation features). The authors do not consider these in their analysis because "strictly related to the structural setting of the area (presence and wavelength of the fold". I don't really understand this statement because each rupture, its splays, its pattern of surface deformation is somehow related to a specific structural pattern (geometry of the fault at depth, segmentation, local arrangement of rock packs, etc). Maybe the authors have in mind the fact that BM and FS are rather related to Coseismic folding (ductile deformation) during earthquake than to Coseismic propagation of the rupture plane to the ground surface? I would suggest to the authors to discuss the way these BM and FS distributed ruptures could be accounted for: this is a critical issue for Italy where thrust-related earthquakes usually occur on blind faults and BM and FS on associated fold are the actual main hazard. In section 3, line 173, the authors seem to mean that there is a direct relation between fold wavelength and location of BM-FS ruptures: this could be a proxy and a way to map and define "Warning or Susceptible zones" to include them in zoning. In line 245-246, the authors write that the*

*"knowledge of the structural setting of the area help in identifying zones potentially susceptible to BM and FS faulting", then why not suggest that such structural features (active folds associated with a thrust) could be defined as "Susceptible Zones"?*

2. *My second comment deals with the definition of metrics and the chosen hypotheses to calculate distances between secondary ruptures and main ruptures. The Figure 1 presents the approach considering a quite simple case, where the "average MF direction" is easy to infer. It is not clear to me how the authors would cope with curved and/or discontinuous ruptures and scarps; at which scale does the average trace is designed? This would change depending on the rupture size (ex. 240 km of Wenchuan vs 15 km of San Fernando). The MS would largely benefit from the inclusion of the rupture maps, so that the reader would understand the authors' method and eventually reproduce the method to improve their work in further steps. Other questions arise for this point: for instance, how the authors measured the distance of secondary ruptures at the tip of the fault, out of the main trace?*

3. *Third comment. The results in terms of statistical outcomes are sufficient to support the conclusions (i.e. definition of three different levels of zones). However, I find the section "Conclusions" look like an Abstract. They should include some perspective or prospective insights, like for instance: - How to take the BM and FS ruptures into account? - Do we need more data to build more robust zoning results? – Would this work or compiled database useful in Probabilistic approach of Fault Displacement Hazard Analysis? - Could future similar developments be applied to other tectonic and permanent deformation features like folding, tilting, extensional/compressional strain (see discussions in ANSI/ANS-2.30-2015 Criteria for Assessing Tectonic Surface Fault Rupture and Deformation at Nuclear Facilities)?*

*SPECIFIC POINTS*

*Replace "associated to" by "associated with" or "related to"*

*Lines 50 51 – Define "main fault" and "secondary faulting"*

*Line 63 - Ambiguous statement: "The AP act defines a minimum distance (: : :) within which critical facilities and structures designed for human occupancy cannot be built". Delete "Critical facilities" because AP Act is only for housing.*

*Line 68 – Please specify which facilities are concerned by "land management" term: only housing, or also lifelines, pipelines, storages or other facilities*

*Line 73 – The fault zones' widths vary at different Levels instead of The fault zones vary at different Levels*

*Line 82 – Please explain what is decided when the "Susceptible Zone" is defined: Avoidance?*

*Line 88 – Rephrase "In general, worldwide the width of the rupture zone : : :"*

*Lines 92 to 95 – Sentence is to be rephrased. Ambiguous sentence: "1: to collect the data from well-studied (: : :) earthquakes" is not exactly was has been done. Instead, it is a compilation of surface maps and displacement observations, not a collection of data (in the field).*

*Lines 117 to 129 - after reading the whole paper, I am not really sure that this description of scarp classification and related section is useful. What is finally used in the conclusions? The authors may earn space here.*

*Lines 135-136 - clarify whether the scarps are described according to the same classification in the references. Or is it a re-interpretation based on drawings, map?*

*Line 142 – The collected data were instead of The collected data was*

*Lines 154 and 155 – The sentence suggests that there is surface geology information in the references: is that the case or is this an assumption?*

*Line 162 - Subsurface data give indication of the location of an anticline on top of the fault: where are located the FS ruptures with respect to this fold?*

*Lines 168-169 – "the distribution of the BM faults for the El Asnam earthquake is very similar to the distribution of extensional ruptures for the San Fernando earthquake": please provide an explanation to this surprising statement. Is there a similar wavelength of associated fault? So no dependence on magnitude (7.3 against 6.6)?*

*Line 182 – Please explain the Kolmogorov-Smirnov test (if you decide to mention it), its specificity and the reason why it has been selected. Otherwise, you can skip this information.*

*Line 186 – Explain why "the 90% probability (: : :) seems to be a reasonable value to cut the outliers": for statistical reasons considering the examined population? Or because its outcome in terms of setback size fits with an a priori?*

*Line 204 – "the data with BM and FS faults are excluded" instead of "the data with BM and FS faults is excluded"*

*Line 209-210 – "the maximum WRZ, including the secondary ruptures away from the main fault, can be up to 200 m or wider": did you explore if there is a relation between this max. width and the earthquake magnitude?*

*Lines 221 to 225 – I would suggest to moderate this with the conclusion coming from the MB and FS analysis. In cases of verified active folds (related to thrusts), this conclusion should be revised, especially for Italy where most of the thrusts are blind (Po Plain) and could cause folding and BM-FS ruptures at the surface as the major hazard.*

*Line 234 – "Assuming that this relation is robust enough" instead of "Assuming that this relation is real" would be better, I think*

*Line 248 – I was wondering if any BM reverse faulting has never been observed in the synclinal axis. This part requires more perspective. How can we account for these ruptures? Any recommendation?*

*Line 282 – This confirms that the avoidance zone should be larger" would be more appropriate than "This suggests that the avoidance zone should be larger" Table 1 - Why 2014 Nagano earthquake rupture has not been included? Figure 1 - There is a different approach for short, intermediate and long secondary ruptures in measuring distance to main fault. Explain please. Caption - Main fault: How is defined main fault? Max. displacement? On the map, this "main fault" runs along the base of the scarp I presume. In complex cases like (d), (e), (f), Main fault trace is traced along the free face or along the more external topographic bulge? Figure 2 – Explain where the Main fault would be mapped on Figure 1 on each case.*

RESPONSE: We would like to thank the reviewer for the appropriate and very useful comments and suggestions. We considered all the comments in the revised version of the manuscript.

The Reviewer's major issues and recommendations have been replied in the following section. All the specific points and minor corrections suggested by the Referee were taken into account.

*MAJOR ISSUES AND RECOMMENDATIONS*

1. *My first comment concerns the Bending-Moment and Flexural Slip ruptures (distributed deformation features). The authors do not consider these in their analysis because "strictly related to the structural setting of the area (presence and wavelength of the fold". I don't really understand this statement because each rupture, its splays, its pattern of surface deformation is somehow related to a specific structural pattern (geometry of the fault at depth, segmentation, local arrangement of rock packs, etc). Maybe the authors have in mind the fact that BM and FS are rather related to Coseismic folding (ductile deformation) during earthquake than to Coseismic propagation of the rupture plane to the ground surface? I would suggest to the authors to discuss the way these BM and FS distributed ruptures could be accounted for: this is a critical issue for Italy where thrust-related earthquakes usually occur on blind faults and BM and FS on associated fold are the actual main hazard. In section 3, line 173, the authors seem to mean that there is a direct relation between fold wavelength and location of BM-FS ruptures: this could be a proxy and a way to map and define "Warning or Susceptible zones" to include them in zoning. In line 245-246, the authors write that the "knowledge of the structural setting of the area help in identifying zones potentially susceptible to BM and FS faulting", then why not suggest that such structural features (active folds associated with a thrust) could be defined as "Susceptible Zones"?*

RESPONSE: We fully understand the Reviewer's criticism. In fact, the most correct approach for considering secondary faulting (particularly ruptures very far from MF) is a difficult task. Basically, our aim in the paper is to distinguish, if feasible, secondary ruptures that occur in a rather "systematic" way compared to the main fault (i.e., only related to the propagation of the main rupture up to the surface), called "simple thrust ruptures", from secondary ruptures that can be affected by structural features that are not systematic (large-scale folds, lithology of folded rocks, …). We discussed this point more deeply and more clearly.

Most importantly, we decided to analyze the data both with and without B-M and F-S distributed ruptures (two different PDFs), and we discussed the results and the possible implications. We also distinguished sympathetic ruptures (Sy). Therefore, we think the paper will be largely improved from this point of view.

2. *My second comment deals with the definition of metrics and the chosen hypotheses to calculate distances between secondary ruptures and main ruptures. The Figure 1 presents the approach considering a quite simple case, where the "average MF direction" is easy to infer. It is not clear to me how the authors would cope with curved and/or discontinuous ruptures and scarps; at which scale does the average trace is designed? This would change depending on the rupture size (ex. 240 km of Wenchuan vs 15 km of San Fernando). The MS would largely benefit from the inclusion of the rupture maps, so that the reader would understand the authors' method and eventually reproduce the method to improve their work in further steps. Other questions arise for this point: for instance, how the authors measured the distance of secondary ruptures at the tip of the fault, out of the main trace?*

RESPONSE: We have added electronic supplementary material, consisting in a summary table with all the measured data and several maps, one for every earthquake, showing the measurement details, including the chosen average strike of the main fault. Moreover, we explained better the method in the text.

In general, we were careful in defining the average strike of the MF and the measurement azimuths, taking into account the variations in strike of the main fault (only first-order, kms-scale strike variations have been considered), and avoiding duplication of measurements. The maps of the auxiliary material will help the reader in judging our choices.

We also added data from the Nagano 2014 earthquake, as suggested.

3. *Third comment. The results in terms of statistical outcomes are sufficient to support the conclusions (i.e. definition of three different levels of zones). However, I find the section "Conclusions" look like an Abstract. They should include some perspective or prospective insights, like for instance: - How to take the BM and FS ruptures into account? - Do we need more data to build more robust zoning results? – Would this work or compiled database useful in Probabilistic approach of Fault Displacement Hazard Analysis? - Could future similar developments be applied to other tectonic and permanent deformation features like folding, tilting, extensional/compressional strain (see discussions in ANSI/ANS-2.30-2015 Criteria for Assessing Tectonic Surface Fault Rupture and Deformation at Nuclear Facilities)?*

RESPONSE: We considered this good suggestion.

[revised manuscript text omitted]

Figure 3 a) Frequency distribution histogram of the distributed ruptures distance (r) from the principal fault rup- ture (PF) for the earthquakes reported in Table 1. The positive and negative values refer to the data on the hang- ing wall and the footwall, respectively; b) Frequency distribution curves of each scarp type excluding those asso- ciated with B-M, F-S and Sy fault ruptures (types h and i of Fig. 2 and sympathetic slip triggered on distant faults); c) Frequency distribution curves of the B-M, F-S and Sy fault ruptures distinguished by earthquakes (the

Sylmar segment extensional zone of the San Fernando 1971 earthquake rupture is included into the B-M fault ruptures).

[Figure]

Figure 4 Cumulative distribution function and probability density function of the rupture distance (r) from the PF

for the hanging wall (a and b, respectively) and the footwall (b and c, respectively) of the PF. Only the scarp types without associated B-M, F-S or sympathetic fault ruptures ("simple thrust" distributed ruptures) were ana- lysed. The 35% probability (HW35) is indicated because it corresponds to sharp drop of the data in the histo- grams.

[Figure]

Figure 5 Cumulative distribution function and probability density function of the rupture distance (r) from the PF

for the hanging wall (a and b, respectively) and the footwall (c and d, respectively) of the PF. All types of distrib- uted ruptures were considered. The 35% probability (HW35) is indicated for comparison with "simple thrust" da- tabase (Fig. 4), but it does not correspond to particular drops of the data in the histograms.

Commento [UW6]: New figure

[Figure]

[Figure]

Figure 6 a) Diagram plotting the total WRZ (WRZtot = WRZ hanging wall + WRZ footwall) against (a) the earthquake magnitude (Mw) and (b) the vertical displacement (VD) on the principal fault.

**Commento [UW7]:** Modified figure